# A multi-isotope model for simulating soil organic carbon cycling in eroding landscapes (WATEM_C v1.0)

Zhengang Wang[1,2], Jianxiu Qiu[1], Kristof Van Oost[2]

[1] Guangdong Provincial Key Laboratory of Urbanization and Geo-simulation, School of geography and planning, Sun Yat-Sen University, Guangzhou 510275, China

[2] Georges Lemaître Center for Earth and Climate Research (TECLIM), Earth and Life Institute, Université catholique de Louvain, 1348 Louvain-la-Neuve, Belgium

*Correspondence to*: Zhengang Wang (wangzhg33@mail.sysu.edu.cn), Jianxiu Qiu (qiujianxiu@mail.sysu.edu.cn)

**Abstract.** There is increasing recognition that lateral soil organic carbon (SOC) fluxes due to erosion have imposed an important impact on the global C cycling. Field and experimental studies have been conducted to investigate this topic. It is useful to have a modelling tool that takes into account various soil properties and has flexible resolution and scale options, so that it can be widely used to study relevant processes and evaluate the effect of soil erosion on SOC cycling. This study presents a model that is capable of simulating SOC cycling in landscapes that are subjected to erosion. It considers all the three C isotopes ($^{12}$C, $^{13}$C and $^{14}$C) with flexible time steps and a detailed vertical solution of the soil profile. The model also represents radionuclide cycling in soils that can assist to constrain the lateral and vertical redistribution of soil particles within landscapes. The model gives a three-dimension representation of soil properties including $^{137}$Cs activity, SOC stock, and $\delta^{13}$C and $\Delta^{14}$C values. Using the same C cycling processes in stable, eroding and depositional areas, our model is able to reproduce the observed spatial and vertical patterns of C contents, $\delta^{13}$C values and $\Delta^{14}$C values. This indicates that at the field scale with similar C decomposition rate, physical soil redistribution is the main cause of the spatial variability of these C metrics.

## 1 Introduction

SOC is the largest organic C pool on land with approximately 1550 Pg C in the upper meter of soil (Lal, 2008). This is about two times of the C in the atmosphere (ca. 760 Pg C). The annual C flux between soil and the atmosphere is ca. 60 Pg C, which makes the atmosphere $CO_2$ sensitive to SOC cycling. SOC stock is a balance between input fluxes and output fluxes, which is controlled by various factors such as soil structure, soil parent material, soil pH, climate and land use and management. Climate is an important controlling factor on SOC cycling as it is closely related to the rate of both C input and decomposition (e.g. Cox et al., 2000; Davidson and Janssens, 2006). The terrestrial net primary production and SOC decomposition rate generally decrease with increasing temperature (Koven et al., 2017). Globally, SOC stock decreases with increasing temperature (Jobbágy and Jackson, 2000). Land use is another important factor because different vegetation supplies SOC to the soil with different rates (Mahowald et al., 2017; Maia et al., 2010). The stable isotopic composition of SOC is affected by factors such as vegetation type ($C_3$ vegetation versus $C_4$ vegetation) and the Suess effect (Tans et al., 1979). Also, SOC can be enriched in $^{13}$C during the processes of SOC degradation due to preferential mineralization of $^{12}$C

(Natelhoffer and Fry, 1988).

Recent studies have shown that lateral soil redistribution by erosion could also impose an important impact on SOC stock and soil-atmosphere C exchange (Chappell et al., 2016; Doetterl et al., 2016). During erosion events, soil aggregates are broken by raindrop and overland flow, which can enhance SOC decomposition (Van Hemelryck et al., 2011). In the eroding region, SOC in the topsoil is removed by erosion resulting in depletion of

SOC. Soil minerals that are moved upwards from below due to soil erosion are added SOC by inputs from plants (Harden et al., 1999). SOC deposited in the depositional settings is buried to depth and the buried SOC is well preserved (Van Oost et al., 2012). This lateral redistribution of SOC and the consequent disturbance of SOC cycling of both eroding and depositional regions result in spatial variability in SOC stocks and properties. It has been found that eroding sites are depleted of SOC compared to the stable sites while depositional sites are enriched

in SOC compared to stable sites in agricultural fields (Li et al., 2007; Van Oost et al., 2005; Yoo et al., 2005). Soil redistribution could lead to differences in SOC stability between eroding and depositional areas. Berhe et al. (2008) found that SOC decomposes at faster rates in eroding areas compared to depositional areas using signatures of $^{14}$C. Wang et al. (2014) reported that SOC mineralization rates in the eroding soil profiles are higher than that of depositional soil profiles from results of laboratory soil incubation. Radioactive C isotopes such as $^{14}$C give

information on the SOC turnover time, and it is a useful tool to investigate long-term SOC cycling (Trumbore, 2009). SOC redistribution has been found to have an effect on SOC radioactive C isotope composition with eroding areas more negative compared to the depositional areas (Berhe et al., 2008).

Apart from the empirical studies mentioned above, various models have been developed to simulate soil erosion and SOC cycling. At the event scale, there are models simulating processes such as rainfall detachment,

sediment entrainment and sediment transport (e.g. Hairsine and Rose, 1992a, b). Some models separate sediments into different grain sizes, and these models are suitable for simulating the size selectivity in erosion and deposition (Nearing, 1989; Van Oost et al., 2004). These models are further modified to simulate the selectivity of SOC in erosion and deposition (Wilken et al., 2017). Models based on USLE (Universal Soil Loss Equation) utilize annual mean precipitation as model input to simulate the long-term soil erosion (Renard et al., 1997). Given that

atmospheric fallouts of $^{137}$Cs are closely adsorbed on soil particles, $^{137}$Cs inventories in soils are widely used to evaluate erosion rates (Gaspar et al., 2013; Quine et al., 1997). Soil erosion models were further added processes of $^{137}$Cs deposition, decay and redistribution associated with soil particles, so that they can be calibrated using observed $^{137}$Cs data (Van Oost et al., 2003).

C turnover models have been developed under the condition of stable landscapes (i.e. free of erosion and

deposition) to explore the effects of climate, land use and soil environment on SOC cycling. The decomposition of C is often represented by a first-order kinetic rate. Because SOC is a complex of different components, it is often represented by various pools with respect to C input and decomposition rates in models such as CENTURY (Parton et al., 1987), ICBM (Andren and Katterer, 1997) and RothC (Coleman and Jenkinson, 1995). C fractions obtained in laboratories have been related to C pools in models and have been used to calibrate model parameters

to investigate the turnover of various C pools (Skjemstad et al., 2004; Wang et al., 2015a; Zimmermann et al., 2007).

These multiple-pool C models were further integrated with soil erosion models to make them applicable at eroding landscapes. For instance, a study adding erosion processes to the CENTURY model has been used to investigate the balance between the lateral SOC loss by erosion and in situ replacement of lost SOC by

photosynthesis at eroding areas, and it has been found that proper management is important to maintain the dynamic replacement of lost C (Harden et al., 1999). At the depositional areas, Wang et al. (2015b) calibrated a profile scale model integrating erosion and SOC cycling processes using observed SOC content and long-term depositional rate, and it was found that sedimentation rate plays an important role in determining burial efficiency of SOC in colluvial settings. Models have also been developed to investigate the relationships between erosion, crop productivity and SOC cycling (Bouchoms et al., 2019). At the field scale, models that combine SOC redistribution by erosion and SOC dynamics are now well able to reproduce the spatial heterogeneity of SOC stock in fields under land uses with eroding areas depleted of SOC and depositional area enriched in SOC (Liu et al., 2003; Rosenbloom et al., 2001; Rosenbloom et al., 2006; Van Oost et al., 2005; Yoo et al., 2005).

Carbon isotopes have also been included in the SOC cycling models to constrain model parameters or explore controlling factors. Baisden et al. (2002) used C and N isotopes to simulate the turnover and transport of SOC along soil depth, and showed that hydrological conditions had an important role in controlling the vertical transport of SOC. Also, a SOC cycling model integrating C isotope discrimination was utilized to explore the effects of SOC decomposition and physical mixing on the formation of the vertical increase of $\delta^{13}C$ values with the soil depth (Acton et al., 2013). Ahrens et al. (2014) used $^{14}C$ signatures to constrain model parameters of a multi-pool SOC model using the Bayesian method, and the model was further applied to quantify the contribution of sorption, dissolved organic carbon transport and microbial interactions in determining the $\Delta^{14}C$ values of soil profiles (Ahrens et al., 2015). Although field studies have identified the effects of soil redistribution on the profile of SOC isotopes (Berhe et al., 2008), relevant models are not developed yet. SOC models including C isotopes applicable in eroding landscapes would be helpful to fully understand the C isotope profiles as well as the spatial variability of SOC isotopic composition at the landscape scale.

Here, we integrate SOC and soil erosion models and present a model tool that is capable of simulating SOC dynamics in an eroding landscape. The objectives of this modelling tool are that (i) it should be a multiple C pool model that is able to represent the complexity of the SOC and to be related to the measurable SOC fractions; (ii) it should include various C isotopes so that it could not only represent these C metrics but also use them to constrain the model; (iii) it should be flexible in terms of spatial and temporal scales so that it would be applicable in various cases regarding spatial and temporal settings.

## 2 Materials and methods

### 2.1 Study sites and field data

The first study site is located in the Belgian Loam Belt. The study area has a temperate climate with an average annual precipitation of 750–800mm and a mean annual temperature of approximately 9.5 °C. Soils in the study area are mainly loess-derived Luvisols with a high silt content (> 70%) and relatively low clay (< 15%) and sand (< 20%) contents (Beuselinck et al., 2000). Arable lands with wheat, maize, sugar beet and potato are the main land use type of the study area. Soil samples were collected from cropland field with rolling topography, where soil cores were taken on plateau (with no erosion), convex slope (with erosion) and concave slope (with deposition) areas. SOC contents were measured with a vario MAX CN Macro Elemental Analyzer (Elementar Analysensysteme GmbH, Germany) while the $\delta^{13}C$ was measured with an ANCA 20-20 GSL mass spectrometer (Sercon Ltd, UK). The inorganic C was removed from soil samples using the HCl-fumigation method proposed by Harris et al. (2001).

This study also used published SOC and $\Delta^{14}C$ data in Berhe et al. (2008) to evaluate the developed model. The

data were collected at Tennessee Valley in Marin County, northern California (37.9 N latitude, 122.6 W longitude).

The climate at Tennessee Valley is Mediterranean, with a mean annual precipitation of 1050 mm and a mean

annual temperature of 14 °C. The dominant vegetation covers in the study area are Mediterranean grasses and a

coastal shrub (Baccharis pilularis, coyote brush). Soils at Tennessee Valley are derived from chert, greenstone,

and greywacke sandstone bedrock of the Franciscan assemblage. Soil profiles at the position of plateau (no erosion),

convex slope (erosion), concave slope (deposition) and valley (deposition) areas were sampled along a slope. C

content was measured with a Carlo Erba elemental analyzer, while the radiocarbon content was measured using

accelerator mass spectrometry (AMS), following the methods of Trumbore et al. (1989).

## 2.2 WATEM_C model

Here we present the WATEM_C model that simulates the redistribution of eroded soil and associated C within the

catchment and its effects on the dynamics of SOC. The soil redistribution by water erosion is based on the WATEM

(Water And Tillage Erosion Model) model (Van Oost et al., 2000) while simulation of C dynamics is based on a

three-pool C model (Wang et al., 2015a). All the three C isotopes ($^{12}C$, $^{13}C$ and $^{14}C$) are included in our model.

Soil advection and diffusion through the soil profile are also included in the model. The model uses flexible time

step and vertical resolution of the soil profile so that it can be applied in various settings.

### 2.2.1 Water erosion
RUSLE (Revised Universal Soil Loss Equation) (Renard et al., 1997) is used to simulate the long-term potential

water erosion ($E_{pot}$; kg m$^{-2}$ yr$^{-1}$):

$$E_{pot} = R * K * L * S * C * P \tag{1}$$

where $R$ is the rainfall erosivity (MJ mm ha$^{-1}$ h$^{-1}$ yr$^{-1}$), $K$ is soil erodibility (kg h MJ$^{-1}$ mm$^{-1}$), $L$ and $S$ are slope

length and steep factors, and $C$ and $P$ are factors for the cover management and support practices.

The local erosion rate is considered to be equal to the potential erosion rate if the potential erosion rate does

not exceed the local transport capacity. The local transport capacity ($T_c$; kg m$^{-1}$ yr$^{-1}$) is calculated as:

$$T_c = k_{tc} * E_{pot} \tag{2}$$

where $k_{tc}$ is the transport capacity coefficient (m). In a grid cell, if its sediment inflow exceeds its local transport

capacity, the amount of material transported through the grid equals to the local transport capacity while the

remainder is deposited in the grid.

A routing algorithm was applied to transfer the mobilized sediments towards the catchment outlet. First, the

grids of the study area were sorted in a descending order based on the DEM. Then, after comparing the local

transport capacity of a grid cell with the incoming sediment and the locally produced sediment (Van Oost et al.,

2000), sediments were routed downslope. Prediction of the flow direction was based on Takken et al. (2001).

The mobilization of SOC or $^{137}Cs$ by erosion ($C_{ero}$, kg m$^{-2}$ yr$^{-1}$) is estimated as:

$$C_{ero} = C_{top} * R_{ero} * ER_{ero} \tag{3}$$

where $C_{top}$ is the content of a carbon isotope or $^{137}Cs$ in the top soil layer (%), $R_{ero}$ is the local erosion rate (kg m$^{-2}$ yr$^{-1}$), $ER_{ero}$ is C enrichment ratio in the eroded sediments that equals to the ratio of C content in the eroded

sediments to that in the source soils.

The deposited SOC or $^{137}$Cs ($C_{depo}$, kg m$^{-2}$ yr$^{-1}$) can be calculated as:

$$C_{depo} = C_{sed} * R_{depo} * ER_{depo} \tag{4}$$

where $C_{sed}$ is the content of a C isotope or $^{137}$Cs in the transported sediments (%), $R_{depo}$ is the local deposition rate (kg m$^{-2}$ yr$^{-1}$), $ER_{depo}$ is C enrichment ratio in the deposited sediments that equals to the ratio of the C content in the deposited sediments to that in the bulk transported sediments reaching the depositional sites.

The enrichment ratios of SOC or $^{137}$Cs in the mobilized sediments at the erosion sites or in the deposited sediments at the deposited sites are found to be closely related to the local erosion or deposition rates (Schiettecatte et al., 2008; Wang et al., 2010). Thus, the enrichment ratios of SOC or $^{137}$Cs in the mobilized and deposited sediments are calculated as:

$$ER_{ero} = a * e^{b*R_{ero}} + 1 \tag{5}$$

$$ER_{depo} = -0.5e^{d*R_{depo}} + 1 \tag{6}$$

where $a$, $b$, and $d$ are coefficients.

### 2.2.2 Soil C turnover

In our model, the three C isotopes ($^{12}$C, $^{13}$C and $^{14}$C) are distinguished. As called in the CENTURY model (Parton et al., 1987), each C isotope is divided into three pools that are referred to as active, slow, and passive pools. The decomposition of these C pools is described using the following differential equations:

$$\frac{d\,^nA(z,t)}{dt} = {}^ni(z) - {}^nk_1r(z)\,{}^nA(z,t) \tag{7}$$

$$\frac{d\,^nS(z,t)}{dt} = h_{AS}\,{}^nk_1r(z)\,{}^nA(z,t) - {}^nk_2r(z)\,{}^nS(z,t) \tag{8}$$

$$\frac{d\,^nP(z,t)}{dt} = h_{AP}\,{}^nk_1r(z)\,{}^nA(z,t) + h_{SP}\,{}^nk_2r(z)\,{}^nS(z,t) - {}^nk_3r(z)\,{}^nP(z,t) \tag{9}$$

where $^nA(z,t)$, $^nS(z,t)$, and $^nP(z,t)$ (%) are the contents of active, slow, and passive pools of C isotope $n$ at depth $z$ (m) and time t (year), respectively; $^ni(z)$ (Mg C yr$^{-1}$) is the input of C isotope n at depth z (m); $h_{AS}$ is the humification coefficients from the active pool to the slow pool, $h_{AP}$ from the active pool to the passive pool, and $h_{SP}$ from the slow pool to the passive pool, respectively; $r(z)$ is a coefficient modifying the variation of C mineralization rate, which denotes the effect of local environmental factors (temperature, humidity, aeration, etc.) at depth $z$ (m); and $^nk_1$, $^nk_2$, and $^nk_3$ (yr$^{-1}$) are the turnover rates at the reference condition (i.e. $r(z) = 1$) of the active, slow, and passive pools of C isotope $n$, respectively.

$^{12}$C is preferentially lost through microbial respiration compared to $^{13}$C and $^{14}$C due to its lower atomic weight (Natelhoffer and Fry, 1988). We used a discrimination ratio to denote the difference in mineralization between isotopes, and thus the decomposition rate of a $^{13}$C pool ($^{13}K_m$, yr$^{-1}$) can be calculated as:

$$^{13}k_m = R_{disc\_13} * {}^{12}k_m \tag{10}$$

where $R_{disc\_13}$ is the discrimination ratio between $^{13}$C and $^{12}$C, $^{12}k_m$ is the decomposition rate of the corresponding $^{12}$C pool.

Similarly, the decomposition rate of a $^{14}$C pool ($^{14}K_m$, yr$^{-1}$) can be calculated as:

$$^{14}k_m = R_{disc\_14} * {}^{12}k_m \tag{11}$$

where $R_{disc\_14}$ is the discrimination ratio between $^{14}C$ and $^{12}C$.

$r$ parameter represents the effect of environmental factors affecting C respiration at a given depth, and it is calculated as:

$$r(z) = r_0 e^{-r_e z} \tag{12}$$

where $r_0$ is the value of $r$ parameter at the top soil layer, and $r_e$ (m$^{-1}$) is an exponential decreasing coefficient .

The input of the C isotopes from plant roots decreases exponentially with depth (Gerwitz and Page, 1974; Van Oost et al., 2005):

$$^{n}i(z) = {}^{n}i_0 e^{-i_e z} \tag{13}$$

where $^{n}i_0$ is the input of C isotope $n$ at the top soil layer (Mg C yr$^{-1}$); $^{n}i(z)$ (Mg C yr$^{-1}$) is the input of C isotope $n$ at depth $z$ (m); and $i_e$ (m$^{-1}$) is an exponential decreasing coefficient.

The $\delta^{13}C$ values are expressed in terms of permil (‰) deviation:

$$\delta^{13}C = \left( \frac{(^{13}C/^{12}C)_{Sample}}{(^{13}C/^{12}C)_{PDB}} - 1 \right) * 1000 \tag{14}$$

where $(^{13}C/^{12}C)_{Sample}$ is the abundance ratio of $^{13}C$ to $^{12}C$ of the soil sample, and $(^{13}C/^{12}C)_{PDB}$ is the ratio of the Pee Dee Belemnite (PDB) as the original standard.

Thus, the $^{13}C$ input can be calculated as:

$$^{13}i = (^{13}C/^{12}C)_{PDB} * \left( 1 + \frac{\delta^{13}C(t)}{1000} \right) * {}^{12}i \tag{15}$$

where $^{13}i$ is the $^{13}C$ input (Mg C ha$^{-1}$ yr$^{-1}$) from plant, $^{12}i$ is the $^{12}C$ input from plant, $\delta^{13}C(t)$ is $\delta^{13}C$ values of C input at time $t$ (yr).

We use the atmospheric $\Delta^{14}C$ record as a proxy for the isotopic ratio of C input via root and leaf litter input (Hua and Barbetti, 2004). In this paper, the following definition of $\Delta^{14}C$ (‰) is used (Stuiver and Polach, 1977):

$$\Delta^{14}C = \left( \frac{(^{14}C/^{12}C)_{Sample}}{A_{ABS}} - 1 \right) * 1000 \tag{16}$$

where $(^{14}C/^{12}C)_{Sample}$ denotes the $^{14}C{:}^{12}C$ ratio of the sample, and $A_{ABS}$ denotes the $^{14}C{:}^{12}C$ ratio of the standard. $A_{ABS}$ is set to be $1.176 * 10^{-12}$ (Karlen et al., 1965; Stuiver, 1980).

The $^{14}C$ input can then be calculated as:

$$^{14}i = A_{ABS} * \left( 1 + \frac{\Delta^{14}CO_2^{ATM}(t)}{1000} \right) * {}^{12}i \tag{17}$$

where $^{14}i$ is $^{14}C$ input (Mg C ha$^{-1}$ yr$^{-1}$) from plant, and the $\Delta^{14}CO_2^{ATM}(t)$ is the atmospheric $\Delta^{14}C$ signals at time $t$ (yr).

### 2.2.3 Soil profile evolution due to erosion

In the model, soil profiles are represented as a series of soil layers with equal depths. Given that C input and SOC decomposition rate are related with soil depth, SOC cycling is simulated in each layer independently. Because
erosion and deposition change the depth of soil profiles, the model updates the depth of soil profiles and the carbon

content of each soil layer every time step. At the eroding locations, soils are removed from the top layer and the soil profile is truncated by the amount of eroded soil. In the meantime, SOC is also lost with the local C enrichment ratio. To keep the soil layer with fixed thickness, soils and associated SOC from soil layers below are incorporated into the upper soil layer at the erosion rate. At the depositional locations, because the top layer is buried by the deposited sediments at the deposition rate, soils and associated SOC are moved downward. For all the soil profiles, the component pools of each C isotope of every layer are updated by homogeneously mixing the component materials every time step.

### 2.2.4 Advection and diffusion
The vertical transport of mineral and organic components of soil is a complex phenomenon driven by a number of distinct mechanisms such as bioturbation (Jagercikova et al., 2017; Johnson et al., 2014), and chemical mobilization (Taylor et al., 2012). We use the advection-diffusion equation to model vertical transport:

$$\frac{dF(z,t)}{dt} = \frac{d^2 K(z)F(z,t)}{dz^2} - \frac{dv(z)F(z,t)}{dz} \tag{18}$$

where $F(z,t)$ is the concentration of a soil constitute (such as a C isotope pool or $^{137}$Cs) at depth $z$ (m) at time $t$ (yr), $v(z)$ (m yr$^{-1}$) is the advection term at depth $z$ (m) and $K(z)$ (m yr$^{-1}$) is a diffusion-type mixing coefficient at depth $z$ (m).

$K$ and $v$ are both depth-dependent and are represented using a sigmoidal scaling function:

$$v(z) = \frac{v_0}{1+e^{(v_d*(z-ct))}} \tag{19}$$

where $v_d$ (m$^{-1}$) is the depth-attenuation of advection, $ct$ (m) is a constant that is set to 0.15 m, and $v_0$ (m yr$^{-1}$) is the value of $v$ at the top soil layer.

$$K(z) = \frac{K_0}{1+e^{(K_d*(z-ct))}} \tag{20}$$

where $K_d$ (m$^{-1}$) is the depth-attenuation of diffusion, and $K_0$ (m yr$^{-1}$) is the value of $K$ at the top soil layer.

### 2.2.5 $^{137}$Cs dynamics
$^{137}$Cs is an artificial nuclear radioactive isotope from nuclear tests and reactor incidents. The $^{137}$Cs in the soil mainly originates from bomb experiments between 1950 and 1970. It falls from the atmosphere primarily in association with precipitation and is rapidly adsorbed to soils by clay materials. As the fallout is well constrained, $^{137}$Cs has been widely used for tracing the movement of soil and sediment particles in erosion studies (Ritchie and McHenry, 1990). The WATEM_C model reads the values of the local $^{137}$Cs fallout. The model then simulates the redistribution of $^{137}$Cs by soil erosion and deposition at the land surface associated with soil particles (Eqs. 1-6). The model also simulates the downward movement of $^{137}$Cs in the soil profile by advection and diffusion (Eq. 18). The decay of $^{137}$Cs (half-life of 30.23 year) is also represented in the model:

$$C_s(z, t + T) = C_s(z,t) * R_p{}^T \tag{21}$$

where $C_s(z, T)$ (%) is $^{137}$Cs content at soil depth $z$ (m) at time $t$ (yr), $R_p$ (equal to 0.977) is the fraction of $^{137}$Cs preserved after the decay of 1 year, and $T$ (yr) is the time step of the model iteration.

### 2.2.6 Model implementation

In order to make the model applicable at various temporal and spatial resolutions, the time step of model iteration and vertical resolution of the soil profile were not fixed, but modifiable as parts of the model input parameters. Given the long-term temporal iteration in SOC cycling processes and the possible large spatial regions where the model may apply, the model was developed using a computation-efficient language (Pascal). The compiled executable file can then be called in other environments such as R (R Development Core Team, 2011) where the

preparation of the input maps is easier. In our model, the default values of the input parameters were given, and at the same time the user are allowed to assign custom values to the input parameters in the R environment when calling the executable file. The description and relevant parameters regarding SOC cycling used in this study are listed in Table 1. For the initialization of the $^{137}$Cs profile, the model first checked if the beginning year of the model simulation was earlier than the beginning of $^{137}$Cs fallout. If earlier, the initial $^{137}$Cs profile was set to be

zero; if not earlier, the model run from the beginning of $^{137}$Cs fallout to the beginning year of the model simulation with relevant parameters values of soil advection and diffusion to generate the initial $^{137}$Cs profile. For the initialization of C profiles, the initial profile of each C pool was set to be the equilibrium C profile under the specific condition to ensure that the initial C profile was realistic for the study site, i.e. the C profile that made the C input equal to the C mineralization, which was determined by model parameters such as C inputs, C turnover

rates and humification coefficients. The model had input parameters of the initial $\delta^{13}$C and $\Delta^{14}$C values of the top and bottom soil layers, and the profiles of $\delta^{13}$C and $\Delta^{14}$C values were then generated by a liner interpolation based on the values of the top and bottom soil layers. The model was then run for a parameter-defined period to get the profiles reach equilibrium condition.

### 2.2.7 Model application

Five model scenarios was tested in this study (Table 2). A set of three scenarios was assumed in order to investigate the effect of advection and diffusion and lateral soil redistribution by erosion on the spatial and vertical distribution of SOC, and $\delta^{13}$C and $\Delta^{14}$C values at the landscape scale. Scenario 1: scenario without advection or diffusion or lateral soil redistribution; Scenario 2: scenario with vertical advection and diffusion but without lateral soil redistribution and Scenario 3: scenario with both advection, diffusion and lateral soil redistribution. In order to

investigate the effect of plant type change and Suess effect on the $\delta^{13}$C values of soil profiles, the model was applied in another set of three scenarios. Given that advection and diffusion is comment in soils, we used the scenario with only advection and diffusion as the reference scenario, i.e. Scenario 2 defined above. The other two scenarios are Scenario 4 with plant type change and Scenario 5 with Suess effect.

   The model was also evaluated using observed C, $\delta^{13}$C and $\Delta^{14}$C soil profiles. Given that an important objective

of this study is to investigate the effects of vertical soil advection and diffusion and lateral soil redistribution on the profiles of C, $\delta^{13}$C and $\Delta^{14}$C, the model was optimized for parameters of $K_0$, $v_0$ and soil redistribution rate, while the other model parameters were set to be realistic values. For the study site in Belgium, the soil redistribution rates varied between soil profiles due to their locations on the hillslope, while the other parameters were set to be the same among soil profiles. For the study site in the USA, the soil redistribution rates varied

between soil profiles. As denoted by Berhe et al. (2008), the grass types vary between slope positions. Also, the top C contents showed great differences between profiles (Figure 1a). Therefore, the C inputs were set to be different between soil profiles. The other parameters were set to be the same among soil profiles for the study site in the USA. The period of erosion was set to be 100 years. The model calibration was performed by comparing

the agreement of the simulations and observations, which included both C contents and C isotopic composition ($\delta^{13}$C values were available at the Belgian study site, while $\Delta^{14}$C data were available at the USA study site). A weight factor was introduced to make sure that both C content and C isotopic composition played equivalent roles in the model calibration.

$$MRMSE = \sqrt{\sum_{j=1}^{Cj}(\frac{SC_j-OC_j}{SD_c})^2 + \sum_{j=1}^{Ij}(\frac{SI_j-OI_j}{SD_I})^2} \tag{22}$$

where MRMSE is the modified root mean square error of the model; $SC_j$ (%) is the simulated C content of sample j; $OC_j$ (%) is the observed C content of sample j; $C_j$ is the number of C content observation, $SD_C$ (%) is the standard deviation of the observed C contents of all the samples; $SI_j$ (‰) is the simulated isotopic composition of sample j, $OI_j$ (‰) is the observed C isotopic composition of sample j; $I_j$ is the number of the observed C isotopic composition of all the samples; $SD_I$ (‰) is the standard deviation of the observed C isotopic composition of all the samples.

In order to quantify the effects of C decomposition, vertical soil advection and diffusion, and lateral soil redistribution on the C, $\delta^{13}$C and $\Delta^{14}$C profiles, comparisons were performed between the reference profiles (i.e. profiles under the condition of Scenario 1) and profiles under a given condition of SOC decomposition, vertical soil advection and diffusion, and lateral soil redistribution using root mean square error (RMSE):

$$RMSE = \sqrt{\sum_{i=1}^{n}(SV_i - OV_i)^2} \tag{23}$$

where $SV_i$ is the simulated value of C content or C isotopic composition of soil layer i; $OV_i$ is the observed value of C content or C isotopic composition of soil layer i; n is the number of soil layers.

The Fourier Amplitude Sensitivity Test (FAST) (Cukier et al., 1973; Cukier et al., 1975) was applied using simulations obtained in a Monte Carlo approach to assess the contribution associated with relevant parameters. The FAST method is based on the analysis of variance (ANOVA) decomposition, which quantifies the relative contribution of only one given parameter to the total variance of the model output. Eight parameter of the model relevant to the C decomposition ($r_0$ and $r_e$), soil advection and diffusion ($K_0$, $K_d$, $v_0$ and $v_d$), and soil redistribution (soil redistribution rate and erosion time) were tested in 10, 000 Monte Carlo scenarios. The values of these parameters were derived from a random distribution within a realistic range ($r_0$: 0.5-1.5; $r_e$: 2.6-4 m$^{-1}$; $K_0$: 0.005-0.1 m yr$^{-1}$; $K_d$: 0.005-0.015 m$^{-1}$; $v_0$: 0.01-0.02 m yr$^{-1}$; $v_d$: 0.005-0.015; soil redistribution rate: -1 - 1 mm yr$^{-1}$; erosion time: 1-100 yr) for each Monte Carlo scenario. No correlations among these input parameters were assumed for sample generation. The FAST test was performed using the MATLAB package developed by Cannavo (2012).

## 3 Results

### 3.1 Model calibration

The optimal parameter values obtained after model calibration were reported in Tables 3 and 4. The model could simulate both the observed C content and C isotopic composition profiles simultaneously well with the MRMSE being 2.26 and 4.46 for the Belgian and the USA study sites, respectively (Figures 1 and 2). The model could not only reproduce the horizontal difference of the C, $\delta^{13}$C and $\Delta^{14}$C profiles between soil profiles well, but that the vertical patterns of these profiles were also well represented by the model, except that the model underestimated the $\Delta^{14}$C values at the top soil layers (Figure 1c).

**3.2 SOC**

Our model is able to reproduce the general pattern of SOC profile of decreasing SOC content with depth in all scenarios despite of rates of advection, diffusion, erosion or deposition (Figure 3). In Scenario 2, higher rates of soil advection and diffusion result in more SOC transferred to the depth, and therefore the difference of SOC content between top layers and bottom layers is smaller under the condition of higher soil advection and diffusion rate compared to SOC profiles of lower advection and diffusion rate (Figure 3b). In Scenario 3, eroding soil profiles contain less SOC compared to the stable soil profiles free of erosion/deposition, while soil profiles at the depositional area are enriched in SOC compare to the stable soil profile (Figure 3c).

**3.3 $\delta^{13}$C values**

In Scenario 1, the $\delta^{13}$C profile shows no variation with depth (Figure 4a). In Scenario 2, the $\delta^{13}$C profile decreases with depth (Figure 4b). The $\delta^{13}$C values of soil profile with higher soil advection and diffusion rates are more negative than that with lower soil advection and diffusion rates (Figure 4b). In Scenario 3, the $\delta^{13}$C values of the eroding profile is less negative than that of the stable soil profile, while soil profiles at the depositional area have more negative $\delta^{13}$C values compared to the stable soil profile (Figure 4c). Our simulation shows that $\delta^{13}$C values increase significantly when the vegetation is converted from $C_3$ vegetation to $C_4$ vegetation (Figure 5). When Suess effect is considered, the $\delta^{13}$C values are lower than that in scenarios that do not consider Suess effect (Figure 5).

**3.4 $\Delta^{14}$C values**

Our model is able to reproduce the general pattern of decreasing $\Delta^{14}$C values with depth in all scenarios despite of rates of soil advection, diffusion, erosion or deposition (Figure 6). In Scenario 2, Soil profiles with higher rates of advection and diffusion have higher $\Delta^{14}$C values compared to profiles with lower vertical transfer rates (Figure 6b). In scenario 3, eroding soil profiles has lower $\Delta^{14}$C values compared to the stable soil profiles, while soil profiles at the depositional area are enriched in $^{14}$C compared to the stable soil profile (Figure 6c).

**3.5 Factors controlling C, $\delta^{13}$C and $\Delta^{14}$C profiles**

C decomposition played the primary role in controlling the C, $\delta^{13}$C and $\Delta^{14}$C profiles with parameter $r_0$ accounting for the major variance of the difference between reference profiles and Monte Carlo scenario profiles (Figure 7). Soil advection and diffusion played a secondary role in controlling the C, $\delta^{13}$C and $\Delta^{14}$C profiles relative to C decomposition, with parameters $K_0$ and $r_0$ contributing more to the variance of the difference between reference profiles and Monte Carlo scenario profiles than parameters $K_d$ and $v_d$. Similarly, Soil redistribution played a relatively secondary role compared to C decomposition, with soil redistribution rate contributing more to the variance of the difference between reference profiles and Monte Carlo scenario profiles than the erosion time. For the C profile, the difference between reference profiles and Monte Carlo scenario profiles was mainly caused by parameter $r_0$, while for the $\delta^{13}$C profile, both $r_0$ and $r_e$ played important roles. However, for the $\Delta^{14}$C profiles, $r_0$, $K_0$, $v_0$ and soil redistribution rate all accounted for more than 15% of the variance of the difference between reference profiles and Monte Carlo scenario profiles.

### 3.6 Spatial variability of soil properties

The model is able to generate a reasonable pattern of soil redistribution with erosion occurring in upland areas and deposition occurring in footslope areas or valleys (Figure 8b). Soil redistribution results in higher [137]Cs inventories in depositional area than eroding area (Figure 8c). The model is also able to generate spatial variability of SOC stock and properties induced by erosion. The depositional area is enriched in SOC compared with eroding area (Figures 8d and 8e). SOC in the depositional area has lower $\delta^{13}C$ values (Figures 8f and 8g) and higher $\Delta^{14}C$ values (Figures 8h and 8i) compared to that in the eroding area.

### 4 Discussion

In Scenario 1, the shape of the SOC profile is determined by the vertical patterns of SOC input and decomposition rates, both of which decrease with depth. The fact that the basic shape of the SOC profile can be well represented in Scenario 1 shows that the pattern of C input and decomposition rates is the primary controlling factor on the SOC profile while other factors such as advection and diffusion, erosion or deposition are relatively secondary (Figure 3a). It is natural that higher rates of advection and diffusion would result in more SOC to be transferred to deep layers (Figure 3b). Given that it is less favorable for SOC to be mineralized in deep layers, the transferred SOC by advection and diffusion to the depth would be better preserved. Simulations in Scenario 2 show that SOC stock in the top 1 m under the condition of high advection and diffusion rate ($K_0$=0.09, $v_0$=0.018) is ca. 14% higher than that under the condition of low advection and diffusion rate ($K_0$=0.05, $v_0$=0.01). Our model can not only reproduce the vertical pattern of SOC distribution in the soil profile, but that it can also reproduce the spatial variability of SOC stock due to soil redistribution. The simulations under Scenario 3 are consistent with observations that soil erosion results in spatial variability of SOC stock (Van Oost et al., 2005; VandenBygaart et al., 2012; Yoo et al., 2005). This model assumes the same C input at both the eroding and depositional areas, and therefore in eroding areas, this C input in combination with the decreased heterotrophic respiration rate caused by the decreased SOC stock by erosion result in replacement of lost SOC at the eroding areas (Harden et al., 1999). Also, SOC buried in the depositional area is partially mineralized over a long period (Van Oost et al., 2012; Wang et al., 2015b). Although offset by the two processes discussed above, soil erosion results in observations that eroding areas are depleted of SOC compared to depositional areas (Van Oost et al., 2005; VandenBygaart et al., 2012; Yoo et al., 2005).In Scenario 1, each soil layer is independent from other soil layers, i.e. there is no mass fluxes between soil layers due to the neglection of advection, diffusion and soil redistribution. In this case, each soil layer has its C input and decomposition rates, which results in the vertical decrease of both [12]C and [13]C with soil depth. The $\delta^{13}C$ value of each soil layer is therefore determined by the discrimination ratio between [13]C and [12]C. If this discrimination ratio is the same between soil layers as implemented in this model, the equilibrium $\delta^{13}C$ profile would be vertically constant (Figure 4a). Due to the fact that the condition of no soil advection and diffusion is not realistic, vertically constant $\delta^{13}C$ profile with depth is rarely reported. When vertical advection and diffusion are considered as in Scenario 2, the transferred SOC from upper layers are isotopically heavier due to degradation compared to the fresh input from plants. This results in an increase of $\delta^{13}C$ values with soil depth (Figure 4b). Our

simulation shows that vertical soil advection and diffusion can be one of the main causes of the widely observed increase of $\delta^{13}C$ profiles with depth (Figure 2). The effect of soil advection and diffusion on the vertial variation of $\delta^{13}C$ values are more profound in soil profiles of high soil advection and diffusion rates. Because erosion and deposition will truncate or bury the original $\delta^{13}C$ profiles, this results in the fact that the eroding soil profile will have higher $\delta^{13}C$ values compared to the stable soil profile while the soil profiles at the depositional sites will have

lower $\delta^{13}C$ values in comparision to the stable soil profile (Figure 4c). This is consistent with the observations made in the croplands in Begium (Figure 2). Also, this discrepancy will be more distinct when the erosion or deposition rates become higher. Our simulation shows that soil redistribution by erosion can also cause spatial variability of $\delta^{13}C$ values on an eroding land.

     Our model is able to reproduce the widely observed decrease of $\Delta^{14}C$ values with depth in soil profiles (Figure

4). Also, the model can capture the signal of bomb carbon with $\Delta^{14}C$ values at the surface layer being positive. In Scenario 1 with no mass fluxes between soil layers, $\Delta^{14}C$ values is mainly a metrics for the turnover rate or residence time of SOC in each layer. The simulated vertical decrease of $\Delta^{14}C$ values is attributed to the vertical variation of environmental conditions that become less favorable for C mineralization. As discussed for $\delta^{13}C$ profiles, at the same depth soil profile of low soil advection and diffusion rate contains more degraded and old

SOC than profile of high soil advetion and diffusion rates, and therefore soil profile of low soil advection and diffusion rate has more negative $\Delta^{14}C$ values (Figure 6b). Similar to $\delta^{13}C$ profiles, erosion and deposition also have a truncation or burial effect of on the $\Delta^{14}C$ profiles and this results in differences of $\Delta^{14}C$ values between disturbed soil profiles and stable soil profiles at the same soil depth. Therefore, the eroding soil profiles have more negative $\Delta^{14}C$ values compared to the stable soil profiles while the profiles at the depositional sites have less

negative $\Delta^{14}C$ values than the stable soil profiles (Figure 6c). Our simulation is consistent with the observation from an eroding hillslope in northen California by Berhe et al. (2008) (Figure 1). The causes of more negative $\Delta^{14}C$ values in eroding soil profiles are mainly attributed to the exposure of old SOC from depth, while the observed less negative $\Delta^{14}C$ values in depositional profiles is due to the burial of young SOC from eroding areas.

     WATEM_C model focuses on the catchment scale, which allows it to account for processes of both erosion

and deposition. It is a spatially distributed model with parcel maps denoting various land use types. Also, it allows accounting for soil conservation measurements, which enables the model to investigate anthropogenic effects (such as land use and management) on erosion and SOC cycling. Compared to previous models, the model presented here is more comprehensive. It includes SOC cycling process and the redistribution of soil and associated SOC by erosion. It is a three-pool C model that discriminates C isotopes ($^{12}C$, $^{13}C$ and $^{14}C$). Thus, it could not only give a

three-dimension representation of C, but also C properties such as $\delta^{13}C$ and $\Delta^{14}C$ values (Figure 8). Our model calibration results show that vertical soil advection and diffusion and lateral soil redistribution could well explain the vertical pattern of C, $\delta^{13}C$ and $\Delta^{14}C$ profiles as well as their spatial variabilities. FAST test shows that C content and C isotopic composition at a given soil depth have different sensitivities to factors such as C decompositon rate, vertical soil advection and diffusion rates and lateral soil redistribution rates (Figure 7). The C content is directly

related to the C decomposition rate, and thus it is mainly controlled by in situ C decompostion rather than vertical soil advection and diffusion and lateral soil redistribution. The effect of C decpmositon on $\delta^{13}C$ and $\Delta^{14}C$ values is not so dominating as on C content, and vetrical and lateral soil redistribution also play important roles in determining the $\delta^{13}C$ and $\Delta^{14}C$ profiles. The default values of most of the parameters was set in the executable file generated in Pascal, but they can be assigned to custom values before running the execable file in R. This allows

the model to be applied in various scenarios of different erosion rates, advection and diffusion rates or vegetation types by setting relevant parameter values. The model is programmed in a computational efficiency langugag (Pascal), which makes it suitable to include more C pools and isotopes. Also, the vertical resolution of the soil profile and the temporal resolution of the model iteration is set to be flexible in our model. The users could modify these parameter based on the requirements of circumstances. The arrangement that the model can be called in R

makes it easier to prepare various input maps and to proceed the output of the model.  However, it requires the users to have experiences in coding in R. The model is designed to simulate only one period with temporally varying inputs on $^{137}$Cs fallout, $^{13}$C and $^{14}$C input. For the cases of temporal variations such as C input or erosion caused by land use change, the current version of the model is not  able to represent these processes.

## 5 Conclusions

This paper presents a model (WATEM_C) that is capable of simulating SOC dynamics on an eroding landscape. It allows tracking the redistribution of soils and associated $^{137}$Cs and SOC within the catchment. The model captures the soil profile evolution due to erosion and deposition. The SOC dynamics was simulated using a three-pool C cycling model. All the three C isotopes ($^{12}$C, $^{13}$C and $^{14}$C) are considered in the model and are discriminated with different cycling rates. The model uses flexible time step and vertical solution of the soil profile. It gives a

three-dimension representation of soil properties such as $^{137}$Cs activity, SOC stock, $\delta^{13}$C values and $\Delta^{14}$C values. Model calibration shows that the model is able to reproduce the observed spatial pattern of the SOC stock that eroding soil profiles are depleted of SOC compared to the stable soil profile while the depositional soil profile is enriched of SOC than the stable soil profile. Our simulation is consistent with the observation that the $\delta^{13}$C values of the eroding profile is less negative than that of the stable soil profile, while soil profiles at the depositional area have more negative $\delta^{13}$C values compared to the stable soil profile. Our model reproduces the observation that

eroding soil profiles has lower $\Delta^{14}$C values compared to the stable soil profiles, while soil profiles at the depositional area are enriched in $^{14}$C compared to the stable soil profile. The fact that the spatial patterns of these SOC metrics can be reproduced using the same C cycling processes indicates that physical soil redistribution is the main cause of these spatial variabilities and that our model captures the most important processes and mechanisms in the SOC cycling on an eroding landscape. FAST test shows that C content is mainly controlled by

in situ C decompostion, while $\delta^{13}$C and $\Delta^{14}$C are also to a large extent affected by processes of vertcial soil advection and diffusion and lateral soil redistribution. We envisage WATEM_C to be a useful tool in simulating the SOC cycling in  eroding landscapes with the wide cover of various soil properties and flexible choices of resolution options and scenario settings.

*Code and data availability.* The source codes is provided through a GitHub repository https://github.com/wangzhg33/Watem_C (last access: 18 August, 2020). A manual on the Watem_C model, data used to conduct model evaluation experiments and examples of using the model are included in the archive files available at http://doi.org/10.5281/zenodo.3988484.

*Author contributions.* All the authors were involved in the design of the model. ZW further developed
       WATEM_C based on the WATEM model by KVO. ZW, JQ and KVO wrote the paper together.

       *Competing interests.* The authors declare that they have no conflict of interest.

*Acknowledgements.* This study was supported by BELSPO (IUAP programme, contract: P7-24) and the
       Natural Science Foundation of China (No. 41871014, 41771216, 41971031).

Tables

**Table 1. Values of parameters on SOC cycling used this study.**

| Parameter | Description | Unit | Scenarios 1-5 | Belgian site | USA site |
|---|---|---|---|---|---|
| $^{12}k_1$ | turnover rates of the active $^{12}C$ pool | $yr^{-1}$ | 2.1 | 2.1 | 2.1 |
| $^{12}k_2$ | turnover rates of the active $^{13}C$ pool | $yr^{-1}$ | 0.03 | 0.03 | 0.03 |
| $^{12}k_3$ | turnover rates of the active $^{14}C$ pool | $yr^{-1}$ | 0.002 | 0.002 | 0.002 |
| $h_{AS}$ | humification coefficients from the active pool to the slow pool | — | 0.12 | 0.12 | 0.12 |
| $h_{AP}$ | humification coefficients from the active pool to the passive pool | — | 0.01 | 0.01 | 0.01 |
| $h_{SP}$ | humification coefficients from the slow pool to the passive pool | — | 0.01 | 0.01 | 0.01 |
| $r_0$ | the r parameter at the top soil layer | — | 1 | 1.035 | 1.78 |
| $i_{root}$ | C input from root | $Mg\ C\ ha^{-1}\ yr^{-1}$ | 2.0 | 2.0 | 19; 14 |
| $i_{resi}$ | C input from leaf litter | $Mg\ C\ ha^{-1}\ yr^{-1}$ | 0 | 0 | 0 |
| $r_e$ | exponential decreasing coefficient of r with depth | $m^{-1}$ | 3.30 | 3.30 | 3.30 |
| $i_e$ | exponential decreasing coefficient for the root C input with depth | $m^{-1}$ | 20 | 20 | 10 |
| $R_{disc\_13}$ | discrimination ratio between $^{13}C$ and $^{12}C$ | — | 0.9977 | 0.9965 | 0.9965 |
| $R_{disc\_14}$ | discrimination ratio between $^{14}C$ and $^{12}C$ | — | 0.996 | 0.996 | 0.996 |

**Table 2. Model scenarios implemented in this study.**

| Scenario | Description | Model implementation |
|---|---|---|
| Scenario 1 | without advection or diffusion or lateral soil redistribution | Set $K_0$ and $v_0$ to be 0;set the period of erosion to be 0 by setting the ending time of erosion to be the same as the starting time of erosion. |
| Scenario 2 | with vertical advection and diffusion but without lateral soil redistribution | Set custom values of $K_0$ and $v_0$; set the period of erosion to be 0 by setting the ending time of erosion to be the same as the starting time of erosion. |
| Scenario 3 | with both advection, diffusion and lateral soil redistribution | Set custom values of $K_0$ and $v_0$; set custom erosion rates and period of erosion. |
| Scenario 4 | with plant type change | Change both the amount and the isotopic composition of C inputs. |
| Scenario 5 | with Suess effect | Change the isotopic composition of C inputs. |


**Table 3. Calibrated optimal parameter values for the Belgian study site. $K_d$ and $v_d$ were set to be 0.01 m⁻¹ in the model calibration. The parameter values on SOC cycling are listed in Table 1. SRR indicates soil redistribution rate.**

| Profiles | $K_0$ (m yr⁻¹) | $v_0$ (m yr⁻¹) | SRR (mm yr⁻¹) |
|---|---|---|---|
| Stable | 0.675 | 0.03 | 0 |
| Erosion | 0.675 | 0.03 | -2.25 |
| Deposition | 0.675 | 0.03 | 3 |

**Table 4. Calibrated optimal parameter values for the USA study site. $K_d$ and $v_d$ were set to be 0.01 m⁻¹ in the model calibration. The parameter values on SOC cycling are listed in Table 1. SRR indicates soil redistribution rate.**

| Profiles | $K_0$ (m yr⁻¹) | $v_0$ (m yr⁻¹) | SRR (mm yr⁻¹) |
|---|---|---|---|
| Stable | 0.0005 | 0.03 | 0 |
| Erosion | 0.0005 | 0.03 | -0.7 |
| Deposition 1 | 0.0005 | 0.03 | 0.8 |
| Deposition 2 | 0.0005 | 0.03 | 2.5 |

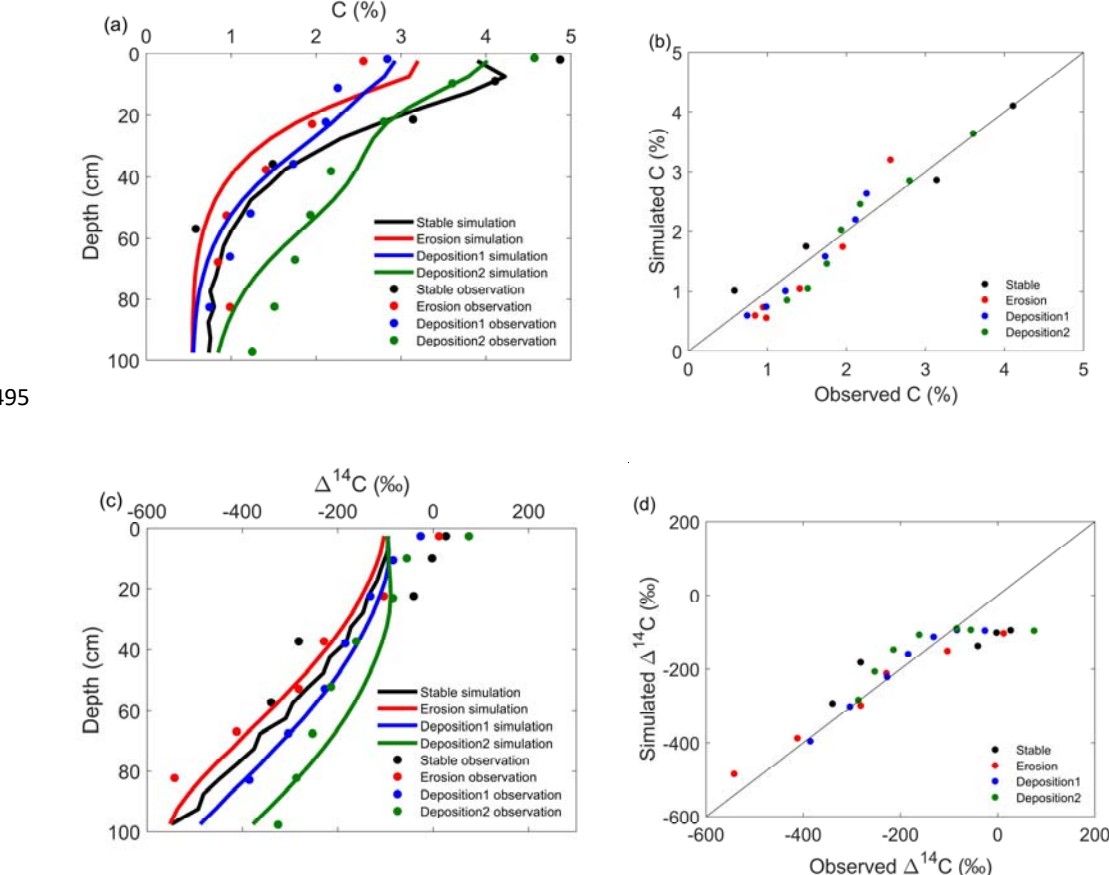

**495**

**Figure 1: Observed and simulated C content and Δ¹⁴C values of stable, erosion, and depositional areas at the USA study site. Observed and simulated C contents in the format of profiles (a) and 1:1 lines (b); observed and simulated Δ¹⁴C values in the format of profiles (c) and 1:1 lines (d).**


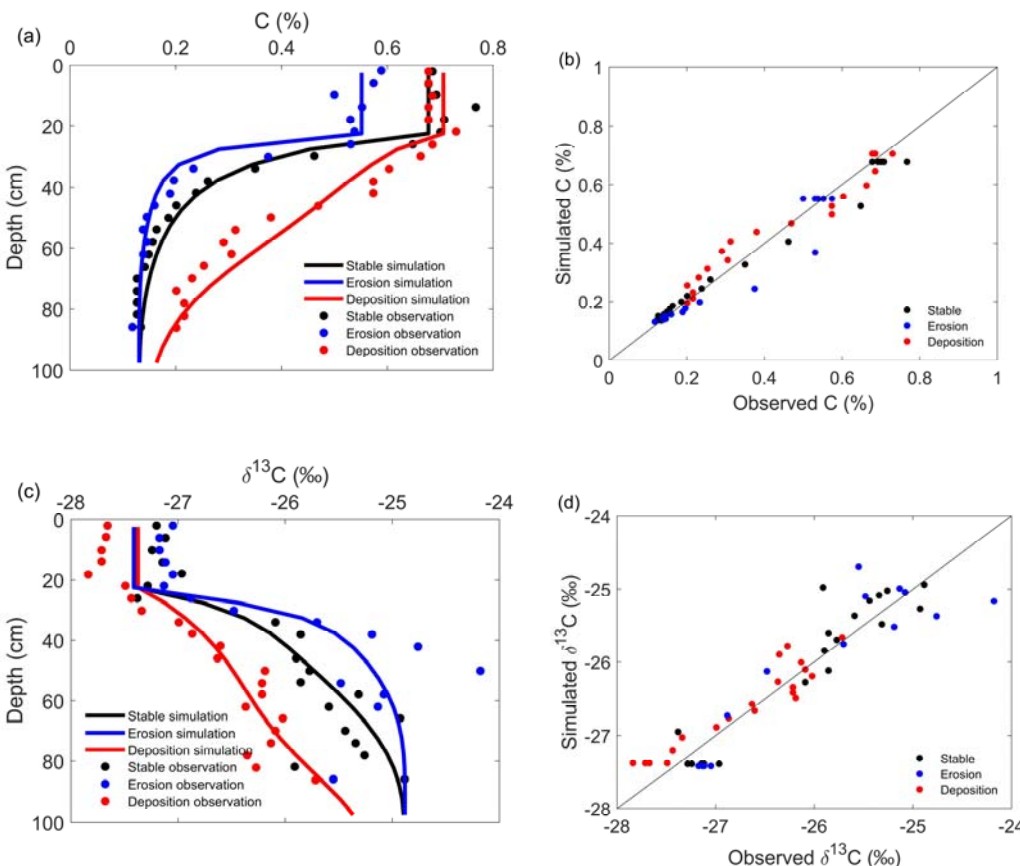

**Figure 2: Observed and simulated C content and δ¹³C values of stable, erosion, and depositional areas at the Belgian study site. Observed and simulated C contents in the format of profiles (a) and 1:1 lines (b); observed and simulated δ¹³C values in the format of profiles (c) and 1:1 lines (d).**


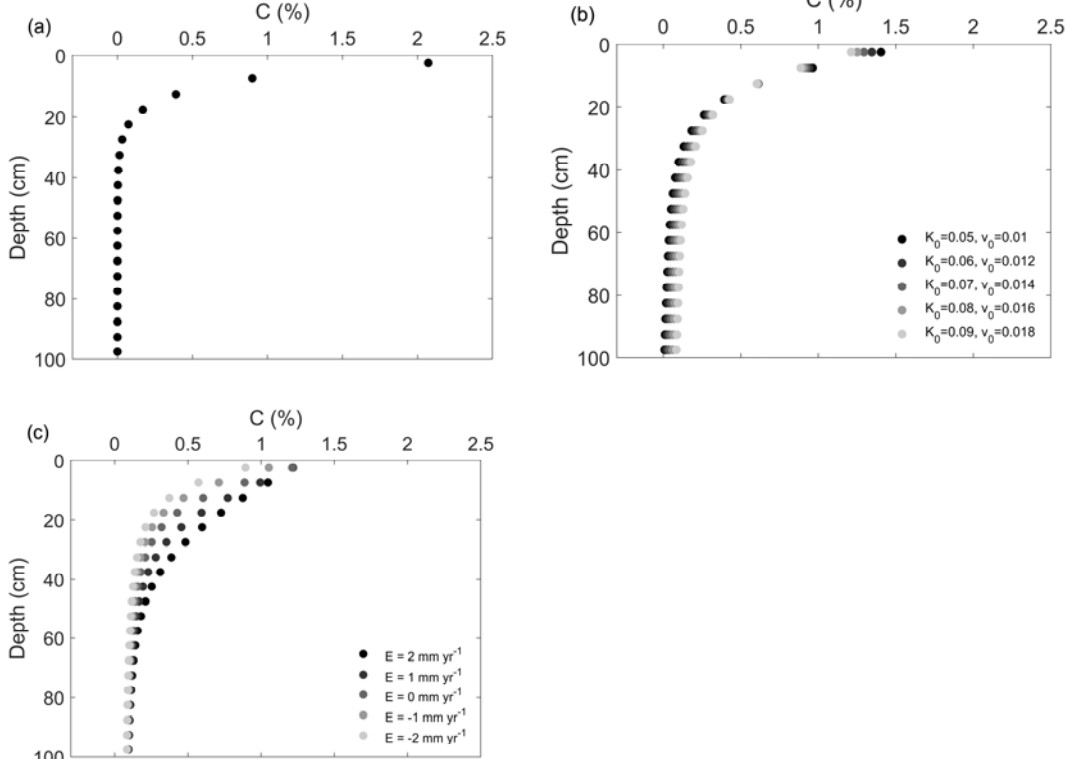

**Figure 3: The simulated C content profiles in (a) Scenario 1, (b) Scenario 2, and (c) Scenario 3. See Section 2.2.7 and Table 2 for the descriptions of scenarios. In b, $K_0$ (m yr$^{-1}$) is the diffusion coefficient at the top soil layer and $v_0$ (m yr$^{-1}$) is the advection term at the top soil layer (Eq. 18). In c, E indicates the soil redistribution rates with negative values indicating erosion and positive values indicating deposition. $K_0$ and $v_0$ were set to be 0.09 m yr$^{-1}$ and 0.018 m yr$^{-1}$, respectively. In b and c, both $K_d$ and $v_d$ (depth- attenuation of diffusion and advection) were set to be 0.01 m$^{-1}$.**


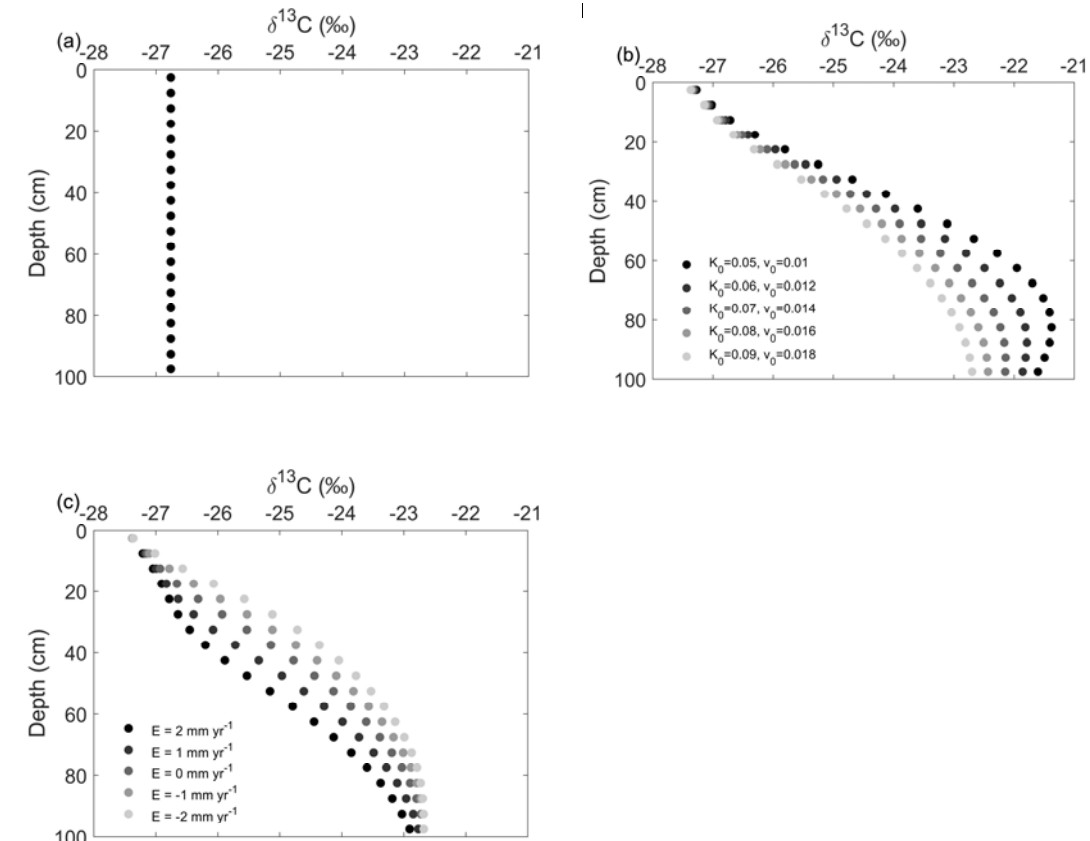


**Figure 4: The simulated δ¹³C profiles in (a) Scenario 1, (b) Scenario 2, and (c) Scenario 3. See Section 2.2.7 and Table 2 for the descriptions of scenarios. In b, $K_0$ (m yr⁻¹) is the diffusion coefficient at the top soil layer and $v_0$ (m yr⁻¹) is the advection term at the top soil layer (Eq. 18). In c, E indicates the soil redistribution rates with negative values indicating erosion and positive values indicating deposition. $K_0$ and $v_0$ were set to be 0.09 m yr⁻¹ and 0.018 m yr⁻¹, respectively. In b and c, both $K_d$ and $v_d$ (depth- attenuation of diffusion and advection) were set to be 0.01 m⁻¹.**



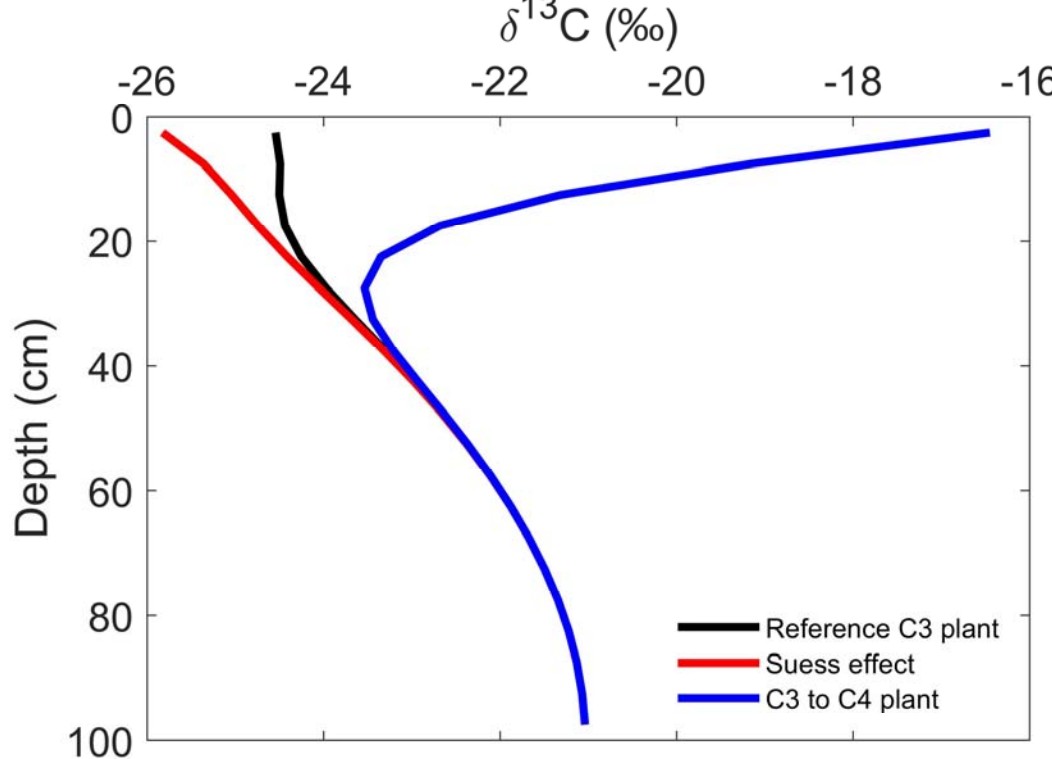

Figure 5: Effects of plant type change and the Suess effect on the $\delta^{13}C$ profiles. In the reference C$_3$ plant scenario, the $\delta^{13}C$ value of C inputs was set to be -26‰; in the Suess effect scenario, the $\delta^{13}C$ value of C inputs decreased from -26‰ to -28.5‰ gradually; in scenario of conversion from C$_3$ plant to C$_4$ plant, the $\delta^{13}C$ value of C inputs was set to be -13‰ after vegetation change.


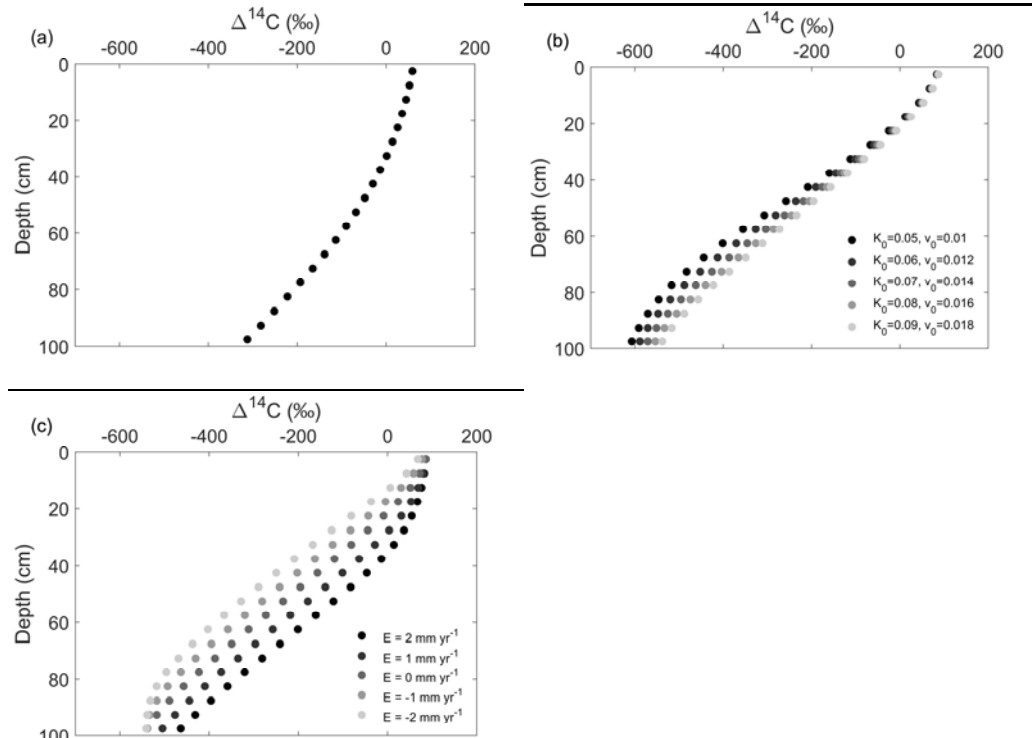

**Figure 6: The simulated $\Delta^{14}C$ profiles in (a) Scenario 1, (b) Scenario 2, and (c) Scenario 3. See Section 2.2.7 and Table 2 for the descriptions of scenarios. In b, $K_0$ (m yr$^{-1}$) is the diffusion coefficient at the top soil layer and $v_0$ (m yr$^{-1}$) is the advection term at the top soil layer (Eq. 18). In c, E indicates the soil redistribution rates with negative values indicating erosion and positive values indicating deposition. $K_0$ and $v_0$ were set to be 0.09 m yr$^{-1}$ and 0.018 m yr$^{-1}$, respectively. In b and c, both $K_d$ and $v_d$ (depth- attenuation of diffusion and advection) were set to be 0.01 m$^{-1}$.**

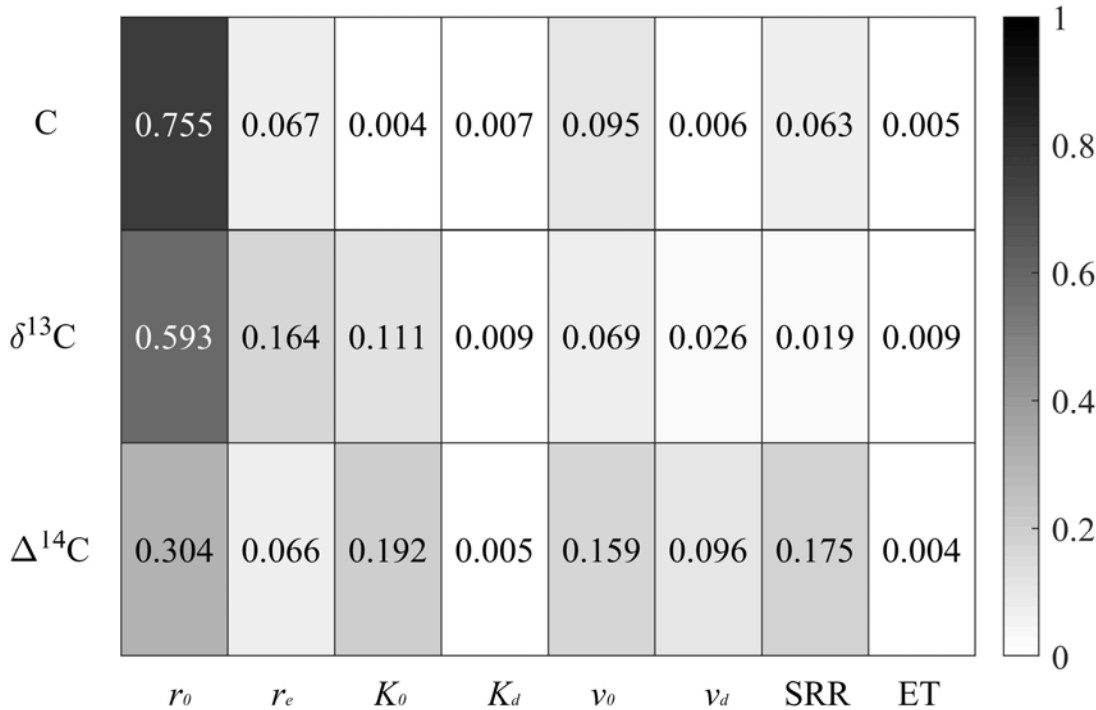

Figure 7: The matrix of proportion of variance of the difference between reference profiles and Monte Carlo scenario profiles caused by model parameters as indicated by the FAST coefficients. SRR indicates soil redistribution rate, and ET indicates erosion time.

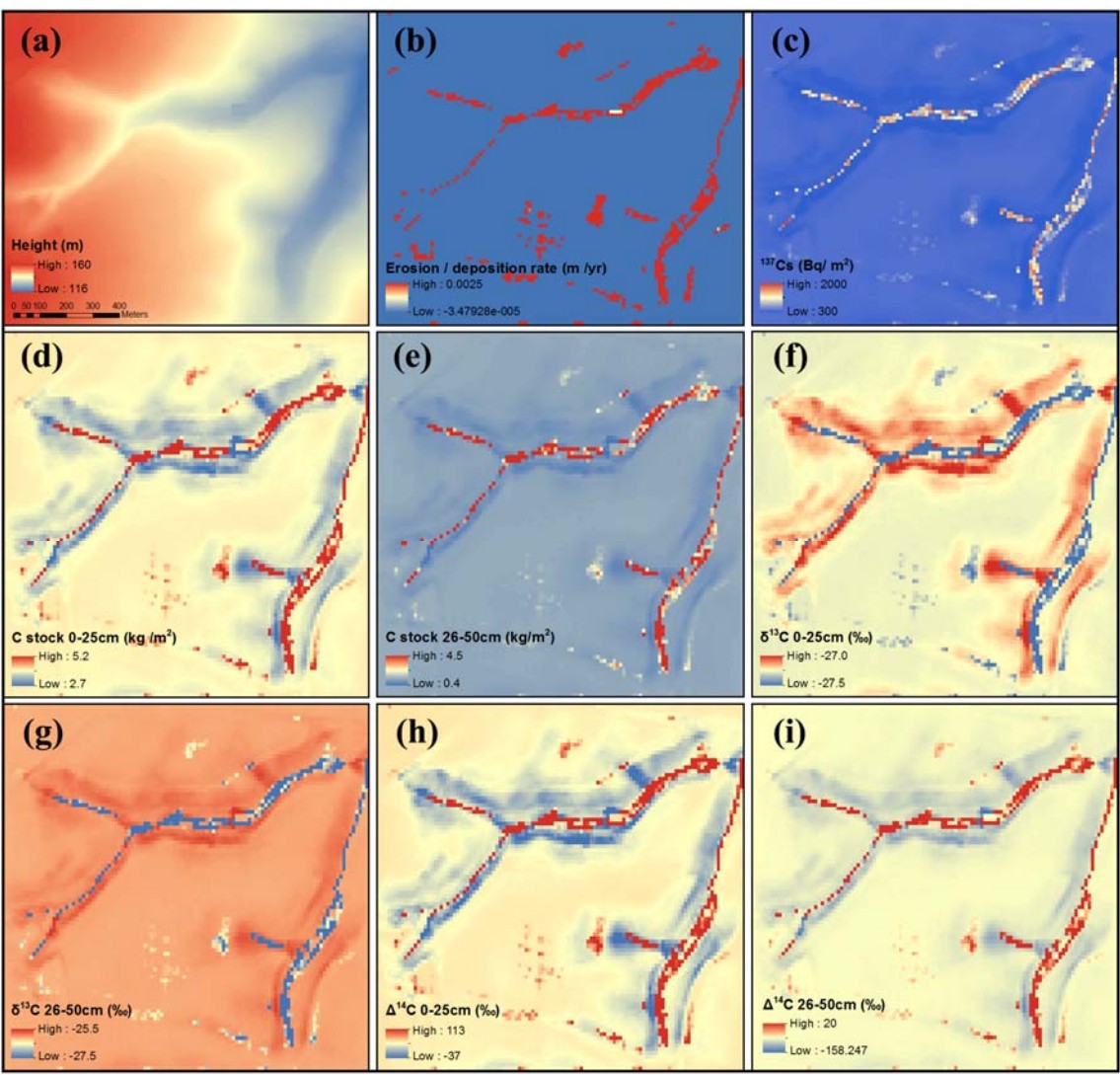

Figure 8: Model simulations of erosion and erosion-induced spatial variability of SOC stock and isotopic compositions. (a) DEM (digital elevation model) of the field, (b) erosion and deposition rates (positive values indicate deposition and negative values indicate erosion), (c) $^{137}$Cs inventory, (d) C stock of topsoil (0–25 cm), (e) C stock of subsoil (26–50 cm), (f) $\delta^{13}$C values of topsoil (0–25 cm), (g) $\delta^{13}$C values of subsoil (26–50 cm), (h) $\Delta^{14}$C values of topsoil (0–25 cm), and (i) $\Delta^{14}$C values of subsoil (26–50 cm).

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
