# Peer review of "A multi-isotope model for simulating soil organic carbon cycling in eroding landscapes (WATEM\_C v1.0)"

_Geoscientific Model Development, 2019_

## Referee Comment (RC1) · Anonymous Referee #1 · 10 Jan 2020

General comments Dear Editors and Authors, This article contributes with a modelling tool (WATEM_C v1.0) for the soil erosion model WATEM, to simulate soil organic carbon cycling in a dynamically eroding landscape. The manuscript is overall fairly well-structured and provides interesting content and results from a novel modelling tool built on integrating soil erosion (WATEM) with SOC models. The use of C isotopes aims to demonstrate that the model can identify sites of erosion and deposition with regards to SOC isotopic enrichment. The text however needs corrections regarding language and grammar. There is also a lack of detail regarding any/the study site used in the model iterations. I recommend major revisions, results- and topic-wise this article would be suitable for publication in Geoscientific Model Development. Below I discuss Specific

and Technical detailed comments and suggested changes.

Specific comments This manuscript provides interesting findings and brings forward a flexible tool that will be useful in research areas covering topics of carbon research and soil erosion. There are some issues that need to be addressed: 1. There are many grammatical and spelling errors in the manuscript, mainly in section 1 Introduction, 3 Results and 4 Discussion. The text is sometimes unclear on what it intends to formulate. I have provided some suggestions in Technical comments. 2. There is a lack of detail regarding any study site that is used to demonstrate the model results. Was this site an actual plot or a computed site? There is a mention of "Belgium" on row 294 in 4 Discussion, but it is not clear how it is related to the findings of this manuscript. If you do have a study site, it should be mentioned, together with basic input parameters, so that other researchers can repeat the simulations. 3. In the introduction, there is a mention of soil erosion and deposition models taking the effect of grain size into consideration, is there any such consideration in your model outside of the scope of the RUSLE components which are inputted into the WATEM model? 4. A suggestion is also to expand the discussion regarding replacement of SOC in eroding areas – for this study did you consider litterfall and vegetation input? For instance, is the vegetation cover heterogeneous in your study area, if so are there any patterns in SOC enrichment that could be connected to vegetation?

Technical comments 1. Consider italicizing the coefficients that are used in the expressions, also in the text for better reading flow. 2. Figures and text make inconsistent use of n-dash and minus signs, suggest streamline for consistency. 3. Row 23: remove "the" in "SOC is the largest organic C pool on the land" 4. Row 25: remove "is" in "atmosphere CO2 sensitive is" 5. Row 37-38: in "During the erosion events, soil aggregates are broken by raindrop and overland flow, which can enhance the SOC decomposition", rewrite to clarify and remove unnecessary "the"s. 6. Row 39-40: "Soil minerals move upwards from below due to soil truncation are added SOC by inputs from plants" – this sentence needs to be rewritten to clarify its meaning, e.g. does

soil truncation force minerals towards the soil surface? And is the SOC added from other sources? 7. Row 41: "SOC deposited in the depositional settings is buried to depth and well preserved", rewrite to clarify 8. Row 43: "SOC tocks", change to stocks 9. Row 43: "It was found that", should more likely be "It has been found" 10. Row 45-47: "Soil redistribution could lead to difference of SOC stability between eroding and depositional areas. Berhe et al. (2008) found that SOC decomposes faster in the eroding areas compared to depositional areas through signatures of radioactive C isotope." These sentences need to be corrected for grammar, and in the last sentence it needs to be clarified that by using radioactive C isotopes it has been found that SOC decomposes at faster rates. Here I also suggest that you write out 14C. 11. Row 49-50: "Radioactive C isotope gives information on the SOC turnover time, and it is a useful tool to investigate long-term SOC cycling (Trumbore, 2009)." Rewrite so that it is grammatically correct. 12. Row 51: "SOC redistribution was found to have an effect", replace was found with "has been found". 13. Row 55-56: a). "Some models separate sediments into different sizes, and these models are suitable for simulations the size selectivity in erosion and deposition" – clarify e.g. by using "into different grain size". Grammar correction: "are suitable for simulating" 14. Row 59-61: "These models were further added processes of 137Cs deposition, decay and redistribution associated with soil particles, so that they can be calibrated using observed 137Cs data (Van Oost et al., 2003). " Here, I think you should give a clear example, to demonstrate why this is relevant. 15. Row 64-65: "Because the SOC is a complex of different components, it is often represented by various pools with respect to C input and decomposition rates in models such as Century" – capitalize and add "CENTURY", remove "the" from "Because the SOC". 16. Row 69-70:" For example, 14C signatures of SOC has been used to constrain parameters of a multiple-pool SOC model using Bayesian method" - check grammar. 17. Row 72-73: Can you give any examples? 18. Row 73-74: Clarify what "at the profile scale" is. Replace "was investigated" with "has been investigated". Also, "they" is not pre-defined, so either introduce the authors of the study you refer to before using "they", or rewrite the sentence into a more generic form 19. Row 75-78:

It is not clear which study these findings are from, clarify 20. Row 82-83: Suggest rewriting this sentence to improve reading flow, would replace "are still lacking" with other expression 21. Row 84: Would use "modelling tool" rather than "model tool" 22. Row 85: perhaps use "eroding landscape" instead of "dynamic landscape" to clarify, or refer to erosion in some other way 23. Row 89: define "scenarios" with e.g. "erosion scenarios" or "erosion settings" to clarify 24. Row 106: space missing "(m).In" 25. Row 126: capitalize CENTURY model 26. Row 132: superscript missing in "ha-1" 27. Row 140: "We used discrimination ratio to", add "a" to "used a discrimination" 28. Row 179: "At the meantime", replace to "In the meantime" or change to other expression 29. Row 182-184: "For all the soil profiles, the component pools of each C isotopes of every layer are updated by homogeneously mixing the component materials every time step." – check grammar 30. Row 188: missing space after ";" 31. Row 200: "137Cs originates from bomb experiments between 1950 and 1970." Very simplified, it is worth clarifying that in the environment 137Cs concentrations are artificial fallout products from nuclear tests and reactor incidents, such as Chernobyl and Fukushima. 32. Row 200: "It falls to the Earth's surface", would use other expression 33. Row 203: "The model reads the values", would remind the reader by clarifying which model 34. Row 211: "the model was develop using", check grammar 35. Row 211: "complied", do you mean "compiled"? 36. Row 276: "replacement of lost at the" lost SOC? 37. Row 280: replace "negelation" 38. Row 286: "from plant." Check grammar 39. Row 288-290: "At the same depth, soil profile of low soil advection and diffusion rate contains more degraded SOC than profile of high soil advetion and diffusion rate, and therefore soil profile of low soil advection and diffusion rate has less negative $\delta$13C values." Check grammar, spelling and clarify the meaning of this sentence 40. Row 293-294: Check grammar and spelling 41. Row 303: Check spelling 42. Row 304-307: "Similar to $\delta$13C profiles, erosion and deposition also have a truncation or burial effect of on the $\Delta$14C profile and this results in the simulation that the eroding soil profiles have more negative $\Delta$14C values compared to the stable soil profile while the profiles at the depositional sites have

less negative $\Delta$14C values than the stable soil profile (Figure 5c)." Check grammar and clarify the meaning 43. Row 318: "three-dimention" check spelling 44. Row 322: "This allows the model to be applied in various scenarios by setting relevant parameter values." Can you clarify which scenarios, e.g. different rates of erosion, different ranges of precipitation or change in vegetation? 45. Row 326: "The arrange", check expression/word 46. Row 330: "reprent", check spelling 47. Row n334: "a 3 pool" be consistent with using words vs. numbers, earlier it has been called a three-pool model 48. Row 339: "while depositional", check grammar "while the depositional" 49. Row 345-346: Check grammar 50. Row 349-350: The link to the code appears to be broken.

Please also note the supplement to this comment:
https://www.geosci-model-dev-discuss.net/gmd-2019-217/gmd-2019-217-RC1-supplement.pdf
* * *

---

## Short Comment (SC1) · 25 Mar 2020

This is an executive editor comment highlighting the ways in which this manuscript is not currently compliant with GMD policy on code and data availability. The issues here must be addressed before a revised manuscript can be accepted for publication:

1. Github URL. Github is an excellent development platform, but it lacks the features required of an archive. GitHub themselves tell authors to use Zenodo for this purpose. The authors should follow the procedure detailed there to archive the exact version of the software used to create the results presented: https://guides. github.com/activities/citable-code/. The resulting Zenodo repositories present the

correct bibliography entries to use.

2. No data identified. The datasets used to conduct the evaluation experiments presented must be cited from the code and data availability section with enough precision to allow a reader to reproduce the work in the manuscript.

3. No configuration, run, or data processing scripts. The configuration files, run scripts and any data processing or analysis scripts used to produce the results presented in the manuscript need to be publicly and persistently archived, and cited from the code and data availability section. As a guide, every file the user would need to reproduce the manuscript should accessible.

Further details on code and data availability requirements are in the GMD model code and data policy: https://www.geoscientific-model-development.net/about/code_ and_data_policy.html. The reasons for the policy and more detail are provided in this editorial: https://doi.org/10.5194/gmd-12-2215-2019.

Use of Github

In addition to the policy compliance issues raised above, I should point out that the authors are currently not really using GitHub in the correct way. Uploading a zip file of the author's installation basically defeats the whole point of revision control. Instead, the git repository should contain the source files and build scripts for the model directly (not in a zip file) as well as the source files for the model documentation and verification tests. Small data sets used for verification could be included, but no other binary files. In particular, including the compiled windows binaries as the authors do makes life difficult for users of their code who will encounter constant conflicts with their own binaries every time they pull updates. If distributing binaries is desirable then this should be accomplished via the appropriate mechanism. See: https://help.github.com/ en/github/administering-a-repository/about-releases

Interactive
comment

---

## Author Comment (AC1) · 5 Apr 2020

Dear editor,

Thanks very much for your comments. We have uploaded the source files and build scripts for the model directed to Github as suggested. For the compiled binaries (WA-TEM_C.ext), we have created a release at the Github, and uploaded the binary to the release as required by the Github. We hope the uploaded files are correct now.

The authors

2019.

---

## Referee Comment (RC2) · Anonymous Referee #2 · 20 May 2020

**comments to the GMD manuscript # A multi-isotope model for simulating soil organic carbon cycling on an eroding land scape**

The authors developed a soil carbon model with coupled processes of decomposition, advection-diffusion and erosion-deposition. The model includes all carbon isotopes and 137Cs. It is a great effort to include all the carbon isotopes in the model. I have several suggestions and concerns below hoping to improve the manuscript.

1.Lack of technical details in the methods section is obvious. To name a few: how did the authors implement plant type change (changes in input, roots and/or else?)? how did the authors test Suess effect in the model? How did the authors simulate spatial

variability? How did the authors determine the initial conditions of the model? What are the depth and depth intervals of the model? More descriptions are necessary for 137Cs dynamics, such as equations and parameter values.

2.A data-model comparison is necessary for model evaluation. I suggest a direct comparison between model outputs and data in figure 7 and figure 8. For example, plot them together.

3.Routine modeling activities such as sensitivity and uncertainty analysis are needed for model evaluations. Relative importance of the three main processes (decomposition, advection-diffusion, and erosion-deposition) need to be shown in some way.

4.An introduction and discussion of progress in model development in carbon isotopes would be very relevant.

5.The results could use some more work. For example, I would not use current figure 1 as the first figure. It is not your central figure. I would first show some figures in model-data comparisons.

A few more comments: 1. What is WATEM_C short for? I did not find its full name throughout the text. 2. Lines 100-105: L and S are slope steep and length factors, or the other way around? 3. Line 211: developED 4. Would a table be helpful for presenting all the modeling scenarios? 5. A, S, and P in equations 7, 8, and 9: are not they carbon concentration (in the unit of for example, gC/m3) instead of carbon content? Because the authors model them with explicit depth. Please clarify. 6. Equation 18: the terms on the right side are supposed to be partial differentials and K(z) is supposed to be inside the second-order differential due to the fact that K changes with z. 7. what are the K values for Fig.1c? and similar issues for other relevant figures. 8. I am surprised to see lack of depth dependence of 13C in Scenario 1. (Fig. 3a). Could the slower decomposition and lower carbon input along depth result in changes in 13C with depth, like 12C and 14C? Please clarify. 9. Fig. 6 is difficult to read. I'd suggest the authors use colored scheme. 10. Line 280: negligence instead of negelation

---

## Author Comment (AC2) · 20 Jul 2020

1. There are many grammatical and spelling errors in the manuscript, mainly in section 1 Introduction, 3 Results and 4 Discussion. The text is sometimes unclear on what it intends to formulate. I have provided some suggestions in Technical comments.

Answer: The suggested revisions have been made in the manuscript with responses for technical comments below. We also check the whole text to correct grammatical and spelling errors if there are any.

2. There is a lack of detail regarding any study site that is used to demonstrate the

model results. Was this site an actual plot or a computed site? There is a mention of "Belgium" on row 294 in 4 Discussion, but it is not clear how it is related to the findings of this manuscript. If you do have a study site, it should be mentioned, together with basic input parameters, so that other researchers can repeat the simulations.

Answer: Data presented in Figure 7 and Figure 8 (of last version) have been used for the model calibration together with C content data at these two study sites. The results derived from model calibration have been presented in Figure 1 and Figure 2 in the revised manuscript. Changes have also been made in Methods (Lines 113-132, Lines 291-317), Results (Lines 336-342), and Discussion (Lines 452-453) sections.

3. In the introduction, there is a mention of soil erosion and deposition models taking the effect of grain size into consideration, is there any such consideration in your model outside of the scope of the RUSLE components which are inputted into the WATEM model?

Answer: As the model generally simulates soil erosion as well as the mobilized SOC using empirical equations as implemented in RUSLE, the model does not focus on the detailed processes regarding grain size which would require parameters on the shear stress, stream power, and flow velocity ect. With respect to the selective transport of SOC, the model introduced an enrichment factor which is related to the erosion or deposition rate (Lines 158 – 174).

4. A suggestion is also to expand the discussion regarding replacement of SOC in eroding areas – for this study did you consider litterfall and vegetation input? For instance, is the vegetation cover heterogeneous in your study area, if so are there any patterns in SOC enrichment that could be connected to vegetation?

Answer: The vegetation cover can be set to be spatially heterogeneous by using a input map with spatial variability, which could cause a spatial variability of C input for the study site. However, the effect of vegetation on the C cycling is not the main focus of this study, and therefore relevant tests were performed in the manuscript. This study

mainly focus on the effects of different erosion-deposition rate on the spatial variability of C content and C isotopic composition. The discussion has been expand regarding the replacement of lateral SOC under the condition of same vegetation (Lines 398-404).

Technical comments

1. Consider italicizing the coefficients that are used in the expressions, also in the text for better reading flow.

Answer: The coefficients have been made italic as suggested.

2. Figures and text make inconsistent use of n-dash and minus signs, suggest streamline for consistency.

Answer: In some places where the n-dash should be used, minus signs wrongly are used in the text and figure. This has been corrected in the revised version of the manuscript.

3. Row 23: remove "the" in "SOC is the largest organic C pool on the land"

Answer: The sentence has been revised to "SOC is the largest organic C pool on land" (Line 26).

4. Row 25: remove "is" in "atmosphere CO2 sensitive is"

Answer: It has been removed (Line 28).

5. Row 37-38: in "During the erosion events, soil aggregates are broken by raindrop and overland flow, which can enhance the SOC decomposition", rewrite to clarify and remove unnecessary "the"s.

Answer: Revisions has been made as suggested (Lines 40-41).

6. Row 39-40: "Soil minerals move upwards from below due to soil truncation are added SOC by inputs from plants" – this sentence needs to be rewritten to clarify its

meaning, e.g. does soil truncation force minerals towards the soil surface? And is the SOC added from other sources?

Answer: The sentence has been rewritten to make it clearer (Line 43).

7. Row 41: "SOC deposited in the depositional settings is buried to depth and well preserved", rewrite to clarify

Answer: It has been rewritten to make it clearer (Line 45).

8. Row 43: "SOC tocks", change to stocks

Answer: It has been corrected (Line 47).

9. Row 43: "It was found that", should more likely be "It has been found"

Answer: It has been revised as suggested (Line 47). Similar revisions have also been made through the text.

10. Row 45-47: "Soil redistribution could lead to difference of SOC stability between eroding and depositional areas. Berhe et al. (2008) found that SOC decomposes faster in the eroding areas compared to depositional areas through signatures of radioactive C isotope." These sentences need to be corrected for grammar, and in the last sentence it needs to be clarified that by using radioactive C isotopes it has been found that SOC decomposes at faster rates. Here I also suggest that you write out 14C.

Answer: The sentence has been corrected as suggested (Lines 50-51).

11. Row 49-50: "Radioactive C isotope gives information on the SOC turnover time, and it is a useful tool to investigate long-term SOC cycling (Trumbore, 2009)." Rewrite so that it is grammatically correct.

Answer: The sentence has been corrected (Line 53).

12. Row 51: "SOC redistribution was found to have an effect", replace was found with "has been found".

Answer: It has been revised as suggested (Line 54).

13. Row 55-56: a). "Some models separate sediments into different sizes, and these models are suitable for simulations the size selectivity in erosion and deposition" – clarify e.g. by using "into different grain size". Grammar correction: "are suitable for simulating"

Answer: It has been corrected as suggested (Line 60).

14. Row 59-61: "These models were further added processes of 137Cs deposition, decay and redistribution associated with soil particles, so that they can be calibrated using observed 137Cs data (Van Oost et al., 2003). " Here, I think you should give a clear example, to demonstrate why this is relevant.

Answer: A short explanation has been added to make the sentence clearer (Line 63-65).

15. Row 64-65: "Because the SOC is a complex of different components, it is often represented by various pools with respect to C input and decomposition rates in models such as Century" – capitalize and add "CENTURY", remove "the" from "Because the SOC".

Answer: It has been revised as suggested (Lines 70, 73).

16. Row 69-70:" For example, 14C signatures of SOC has been used to constrain parameters of a multiple-pool SOC model using Bayesian method" - check grammar.

Answer: This sentence has been removed in the revised manuscript. As suggested by the other reviewer, a paragraph on the progress of model developed using C isotopes has been added (Lines 91-102).

17. Row 72-73: Can you give any examples?

Answer: Actually, this sentence is a summary of the paragraph, and the following sentences are examples. We have revised the paragraph to make this clear (Line 79).

18. Row 73-74: Clarify what "at the profile scale" is. Replace "was investigated" with "has been investigated". Also, "they" is not pre-defined, so either introduce the authors of the study you refer to before using "they", or rewrite the sentence into a more generic form

Answer: This sentence has been rewritten to make it clear (Lines 79-82).

19. Row 75-78: It is not clear which study these findings are from, clarify

Answer: The sentence has been rewritten to avoid misunderstanding (Line 83).

20. Row 82-83: Suggest rewriting this sentence to improve reading flow, would replace "are still lacking" with other expression

Answer: This sentence has been removed in the revised version of the manuscript (Lines 103-105).

21. Row 84: Would use "modelling tool" rather than "model tool"

Answer: It has been revised as suggested (Line 107).

22. Row 85: perhaps use "eroding landscape" instead of "dynamic landscape" to clarify, or refer to erosion in some other way

Answer: It has been revised as suggested (Line 107).

23. Row 89: define "scenarios" with e.g. "erosion scenarios" or "erosion settings" to clarify

Answer: It has been clarified by "cases regarding spatial and temporal settings" (Line 111).

24. Row 106: space missing "(m).In"

Answer: A space has been added (Line 149).

25. Row 126: capitalize CENTURY model

Answer: It has been revised as suggested (Line 176).

26. Row 132: superscript missing in "ha-1"

Answer: It has been replaced to % as suggested by the other reviewer (Line 183).

27. Row 140: "We used discrimination ratio to", add "a" to "used a discrimination"

Answer: "a" has been added (Line 190).

28. Row 179: "At the meantime", replace to "In the meantime" or change to other expression

Answer: "At the meantime" has been replaced with "In the meantime" (Line 229).

29. Row 182-184: "For all the soil profiles, the component pools of each C isotopes of every layer are updated by homogeneously mixing the component materials every time step." – check grammar

Answer: We have change "isotopes" to "isotope" (Line 233).

30. Row 188: missing space after ";"

Answer: This issue has been solved in the reference manage tool.

31. Row 200: "137Cs originates from bomb experiments between 1950 and 1970." Very simplified, it is worth clarifying that in the environment 137Cs concentrations are artificial fallout products from nuclear tests and reactor incidents, such as Chernobyl and Fukushima.

Answer: It has been revised to make it more detailed as suggested (Line 250).

32. Row 200: "It falls to the Earth's surface", would use other expression

Answer: It has been revised (Line 251).

33. Row 203: "The model reads the values", would remind the reader by clarifying which model

Answer: The name of the model has been added (Line 254).

34. Row 211: "the model was develop using", check grammar

Answer: "develop" has been changed to "developed" (Line 265).

35. Row 211: "complied", do you mean "compiled"?

Answer: "complied" has been replaced with "compiled" (Line 265).

36. Row 276: "replacement of lost at the" lost SOC?

Answer: The missing "SOC" has been added (Line 400).

37. Row 280: replace "negelation"

Answer: "negelation" has been replaced with "neglection" (Line 408).

38. Row 286: "from plant." Check grammar

Answer: It has been changed to "from plants" (Line 415).

39. Row 288-290: "At the same depth, soil profile of low soil advection and diffusion rate contains more degraded SOC than profile of high soil advetion and diffusion rate, and therefore soil profile of low soil advection and diffusion rate has less negative $\delta$13C values." Check grammar, spelling and clarify the meaning of this sentence

Answer: We have rewritten the sentence to make it clearer (Lines 417-418).

40. Row 293-294: Check grammar and spelling

Answer: It has been revised by using the plurals of "observation" and "cropland" (Line 424).

41. Row 303: Check spelling

Answer: "rate" has been replaced with "rates" (Line 433).

42. Row 304-307: "Similar to $\delta$13C profiles, erosion and deposition also have a trun-

cation or burial effect of on the $\Delta14C$ profile and this results in the simulation that the eroding soil profiles have more negative $\Delta14C$ values compared to the stable soil profile while the profiles at the depositional sites have less negative $\Delta14C$ values than the stable soil profile (Figure 5c)." Check grammar and clarify the meaning

Answer: We have revised the sentence to make it clearer (Lines 435-436).

43. Row 318: "three-dimention" check spelling

Answer: It has been changed to "three-dimension" (Line 449).

44. Row 322: "This allows the model to be applied in various scenarios by setting relevant parameter values." Can you clarify which scenarios, e.g. different rates of erosion, different ranges of precipitation or change in vegetation?

Answer: The types of scenarios have been clarified (Lines 461-462).

45. Row 326: "The arrange", check expression/word

Answer: It has been changed to "The arrangement" (Line 465).

46. Row 330: "reprent", check spelling

Answer: It has been replaced with "represent" (Line 469).

47. Row n334: "a 3 pool" be consistent with using words vs. numbers, earlier it has been called a three-pool model

Answer: It has been placed with "a three-pool" (Line 473).

48. Row 339: "while depositional", check grammar "while the depositional"

Answer: "the" has been added (Line 478).

49. Row 345-346: Check grammar

Answer: "causes" has been changed to "cause" (Line 485).

50. Row 349-350: The link to the code appears to be broken.

Answer: The files have been updated as suggested by the editor.
* * *

---

## Author Comment (AC3) · 20 Jul 2020

The authors developed a soil carbon model with coupled processes of decomposition, advection-diffusion and erosion-deposition. The model includes all carbon isotopes and 137Cs. It is a great effort to include all the carbon isotopes in the model. I have several suggestions and concerns below hoping to improve the manuscript.

1.Lack of technical details in the methods section is obvious. To name a few: how did the authors implement plant type change (changes in input, roots and/or else?) how did the authors test Suess effect in the model?

Answer: A table on the description of various scenarios and how they were performed has been added (Table 2).

How did the authors simulate spatial variability?

Answer: The spatial variability is related to soil redistribution at the landscape scale. A paragraph has been added to describe in detail how the routing of runoff and soil particles are simulated in the model (Lines 142-145).

How did the authors determine the initial conditions of the model?

Answer: A detailed description of the procedure to estimate/set the initial profiles of 137Cs, C pools and C isotopic compositions has been added (Lines 270-280). We believe that this now provides sufficient information for the readers.

What are the depth and depth intervals of the model?

Answer: These are defined by parameters in the model code. Annotations have been added in the R script (reference_scenario.R) to explain the meanings of variables used in the codes. The R script file has been updated.

More descriptions are necessary for 137Cs dynamics, such as equations and parameter values.

Answer: Section 2.2.1 has been revised to include the lateral fluxes of 137Cs due to soil erosion. The decay of 137Cs (Eq. 21) has been added in section 2.2.5 (Lines 258-260).

2.A data-model comparison is necessary for model evaluation. I suggest a direct comparison between model outputs and data in figure 7 and figure 8. For example, plot them together.

Answer: Data presented in Figure 7 and Figure 8 (of last version) have been used for the model calibration together with C content data at these two study sites. The results derived from model calibration have been presented in Figure 1 and Figure 2 in the

revised manuscript. Changes have also been made in Methods (Lines 113-132, Lines 291-317), Results (Lines 336-342), and Discussion (Lines 452-453) sections.

3.Routine modeling activities such as sensitivity and uncertainty analysis are needed for model evaluations. Relative importance of the three main processes (decomposition, advection-diffusion, and erosion-deposition) need to be shown in some way.

Answer: Thank you for this valuable suggestion. In response to this comment, the Fourier Amplitude Sensitivity Test (FAST) has been applied to the model to explore the importance of C decomposition, advection-diffusion and erosion-deposition in controlling C, $\delta$13C and $\Delta$14C profiles. The results is presented in Figure 7. Changes have also been made in Methods (Lines 318-327), Results (Lines 366-377), Discussion (Lines 454-459) and Conclusion (Lines 486-488) sections.

4.An introduction and discussion of progress in model development in carbon isotopes would be very relevant.

Answer: A paragraph on a review of progress in model development in carbon isotopes have been added (Lines 91-102).

5.The results could use some more work. For example, I would not use current figure 1 as the first figure. It is not your central figure. I would first show some figures in model-data comparisons.

Answer: Figure 1 has been removed from the revised manuscript. Figures on model-data (Figures 1 and 2) has been shown first as suggested (see the reply to Point 2 above).

A few more comments:

1. What is WATEM_C short for? I did not find its full name throughout the text.

Answer: The abbreviation has been explained (Line 136).

2. Lines 100-105: L and S are slope steep and length factors, or the other way around?

Answer: It is the other way around, and it has been corrected (Line 145).

3. Line 211: developED

Answer: It has been corrected as suggested (Line 265).

4. Would a table be helpful for presenting all the modeling scenarios?

Answer: A table including descriptions and implementation of model scenarios has been added (Table 2). Relevant revisions has also been done in the text (Line 282).

5. A, S, and P in equations 7, 8, and 9: are not they carbon concentration (in the unit of for example, gC/m3) instead of carbon content? Because the authors model them with explicit depth. Please clarify.

Answer: The reviewer is correct that A, S and P should be the content of various C pools. Similarly, the unit of C input into a given depth has been changed from Mg C ha -1 yr-1 to Mg C yr-1 (Lines 182, and 183).

6. Equation18: the terms on the right side are supposed to be partial differentials and K(z) is supposed to be inside the second-order differential due to the fact that K changes with z.

Answer: Eq. 18 has been revised as suggested.

7. what are the K values for Fig.1c? and similar issues for other relevant figures.

Answer: Values of relevant parameters have been added in the figures (Figures 3-6 in the revised manuscript). We also extended Table 1 and added Tables 2 and 3 to display more information on parameter values of the model.

8.I am surprised to see lack of depth dependence of 13C in Scenario 1. (Fig. 3a). Could the slower decomposition and lower carbon input along depth result in changes in 13C with depth, like 12C and 14C? Please clarify.

Answer: Figure 6a (in the revised manuscript) shows the vertical variation of $\delta$13C

values rather 13C content. In Scenario 1, slower decomposition and lower carbon input along depth result in changes in 13C with depth, but 12C has similar changes with soil depth, and therefore the 13C composition ($\delta$13C) does not change with soil depth. A sentence has been added to explain this (Line 409).

9. Fig. 6 is difficult to read. I'd suggest the authors use colored scheme.

Answer: The figure has been changed to color maps (Figure 8 in the revised version).

10. Line 280: negligence instead of negelation

Answer: It has been replaced with neglection (Line 408).

---

## Author Response (AR1)

Dear editor,

Thank you very much for your review on our manuscript. Below is our response to the reviewers' comments. They are referred to the line numbers in **the manuscript with revisions marked,** which was uploaded as a supplementary material.

Yours sincerely,

The authors

**Reviewer 1:**

**Comments from reviewer**

Comments on the manuscript "A multi-isotope model for simulating soil organic carbon cycling on an eroding landscape (WATEM_C v1.0)" to *Geoscientific Model Development Discussions* by Wang & Van Oost, 2019.

**General comments**

Dear Editors and Authors,

This article contributes with a modelling tool (WATEM_C v1.0) for the soil erosion model WATEM, to simulate soil organic carbon cycling in a dynamically eroding landscape. The manuscript is overall fairly well-structured and provides interesting content and results from a novel modelling tool built on integrating soil erosion (WATEM) with SOC models. The use of C isotopes aims to demonstrate that the model can identify sites of erosion and deposition with regards to SOC isotopic enrichment. The text however needs corrections regarding language and grammar. There is also a lack of detail regarding any/the study site used in the model iterations.

I recommend major revisions, results- and topic-wise this article would be suitable for publication in Geoscientific Model Development.

Below I discuss my detailed comments and suggested changes.

**Specific comments**

This manuscript provides interesting findings and brings forward a flexible tool that will be useful in research areas covering topics of carbon research and soil erosion. There are some issues that need to be addressed:

1. There are many grammatical and spelling errors in the manuscript, mainly in section *1 Introduction, 3 Results* and *4 Discussion*. The text is sometimes unclear on what it intends to formulate. I have provided some suggestions in Technical comments.

**The suggested revisions have been made in the manuscript with responses for technical comments below. We also check the whole text to correct grammatical and spelling errors if there are any.**

2. There is a lack of detail regarding any study site that is used to demonstrate the model results. Was this site an actual plot or a computed site? There is a mention of "Belgium" on row 294 in *4 Discussion*, but it is not clear how it is related to the findings of this manuscript. If you do have a study site, it should be mentioned, together with basic input parameters, so that other researchers can repeat the simulations.

**Data presented in Figure 7 and Figure 8 (of last version) have been used for the model calibration together with C content data at these two study sites. The results derived from model calibration have been presented in Figure 1 and Figure 2 in the revised manuscript. Changes have also been made in Methods (Lines 113-132, Lines 291-317), Results (Lines 336-342), and Discussion (Lines 452-453) sections.**

3. In the introduction, there is a mention of soil erosion and deposition models taking the effect of grain size into consideration, is there any such consideration in your model outside of the scope of the RUSLE components which are inputted into the WATEM model?

**As the model generally simulates soil erosion as well as the mobilized SOC using empirical equations as implemented in RUSLE, the model does not focus on the detailed processes regarding grain size which would require parameters on the shear stress, stream power, and flow velocity ect. With respect to the selective transport of SOC, the model introduced an enrichment factor which is related to the erosion or deposition rate (Lines 158 – 174).**

4. A suggestion is also to expand the discussion regarding replacement of SOC in eroding areas – for this study did you consider litterfall and vegetation input? For instance, is the vegetation cover heterogeneous in your study area, if so are there any patterns in SOC enrichment that could be connected to vegetation?

**The vegetation cover can be set to be spatially heterogeneous by using a input map with spatial variability, which could cause a spatial variability of C input for the study site. However, the effect of vegetation on the C cycling is not the main focus of this study, and therefore relevant tests were performed in the manuscript. This study mainly focus on the effects of different erosion-deposition rate on the spatial variability of C content and C isotopic composition. The discussion has been expand regarding the replacement of lateral SOC under the condition of same vegetation (Lines 398-404).**

**Technical comments**

1. Consider italicizing the coefficients that are used in the expressions, also in the text for better reading flow.

**The coefficients have been made italic as suggested.**

2. Figures and text make inconsistent use of *n*-dash and minus signs, suggest streamline for consistency.

**In some places where the n-dash should be used, minus signs wrongly are used in the text and figure. This has been corrected in the revised version of the manuscript.**

3. Row 23: remove "the" in "SOC is the largest organic C pool on the land"

**The sentence has been revised to "SOC is the largest organic C pool on land" (Line 26).**

4. Row 25: remove "is" in "atmosphere $CO_2$ sensitive is"

**It has been removed (Line 28).**

5. Row 37-38: in "During the erosion events, soil aggregates are broken by raindrop and overland flow, which can enhance the SOC decomposition", rewrite to clarify and remove unnecessary "the"s.

**Revisions has been made as suggested (Lines 40-41).**

6. Row 39-40: "Soil minerals move upwards from below due to soil truncation are added SOC by inputs from plants" – this sentence needs to be rewritten to clarify its meaning, e.g. does soil truncation force minerals towards the soil surface? And is the SOC added from other sources?

**The sentence has been rewritten to make it clearer (Line 43).**

7. Row 41: "SOC deposited in the depositional settings is buried to depth and well preserved", rewrite to clarify

**It has been rewritten to make it clearer (Line 45).**

8. Row 43: "SOC tocks", change to stocks

**It has been corrected (Line 47).**

9. Row 43: "It was found that", should more likely be "It has been found"

**It has been revised as suggested (Line 47). Similar revisions have also been made through the text.**

10. Row 45-47: "Soil redistribution could lead to difference of SOC stability between eroding and depositional areas. Berhe et al. (2008) found that SOC decomposes faster in the eroding areas compared to depositional areas through signatures of radioactive C isotope." These sentences need to be corrected for grammar, and in the last sentence it needs to be clarified that by using radioactive C isotopes it has been found that SOC decomposes at faster rates. Here I also suggest that you write out $^{14}$C.

**The sentence has been corrected as suggested (Lines 50-51).**

11. Row 49-50: "Radioactive C isotope gives information on the SOC turnover time, and it is a useful tool to investigate long-term SOC cycling (Trumbore, 2009)." Rewrite so that it is grammatically correct.

**The sentence has been corrected (Line 53).**

12. Row 51: "SOC redistribution was found to have an effect", replace was found with "has been found".

**It has been revised as suggested (Line 54).**

13. Row 55-56: a). "Some models separate sediments into different sizes, and these models are suitable for simulations the size selectivity in erosion and deposition" – clarify e.g. by using "into different grain size". Grammar correction: "are suitable for simulating"

**It has been corrected as suggested (Line 60).**

14. Row 59-61: "These models were further added processes of $^{137}$Cs deposition, decay and redistribution associated with soil particles, so that they can be calibrated using observed $^{137}$Cs data (Van Oost et al., 2003). " Here, I think you should give a clear example, to demonstrate why this is relevant.

**A short explanation has been added to make the sentence clearer (Line 63-65).**

15. Row 64-65: "Because the SOC is a complex of different components, it is often represented by various pools with respect to C input and decomposition rates in models such as Century" – capitalize and add "CENTURY", remove "the" from "Because the SOC".

**It has been revised as suggested (Lines 70, 73).**

16. Row 69-70:" For example, $^{14}$C signatures of SOC has been used to constrain parameters of a multiple-pool SOC model using Bayesian method" - check grammar.

**This sentence has been removed in the revised manuscript. As suggested by the other reviewer, a paragraph on the progress of model developed using C isotopes has been added (Lines 91-102).**

17. Row 72-73: Can you give any examples?

**Actually, this sentence is a summary of the paragraph, and the following sentences are examples. We have revised the paragraph to make this clear (Line 79).**

18. Row 73-74: Clarify what "at the profile scale" is. Replace "was investigated" with "has been investigated". Also, "they" is not pre-defined, so either introduce the authors of the study you refer to before using "they", or rewrite the sentence into a more generic form

**This sentence has been rewritten to make it clear (Lines 79-82).**

19. Row 75-78: It is not clear which study these findings are from, clarify

**The sentence has been rewritten to avoid misunderstanding (Line 83).**

20. Row 82-83: Suggest rewriting this sentence to improve reading flow, would replace "are still lacking" with other expression

**This sentence has been removed in the revised version of the manuscript (Lines 103-105).**

21. Row 84: Would use "modelling tool" rather than "model tool"

**It has been revised as suggested (Line 107).**

22. Row 85: perhaps use "eroding landscape" instead of "dynamic landscape" to clarify, or refer to erosion in some other way

**It has been revised as suggested (Line 107).**

23. Row 89: define "scenarios" with e.g. "erosion scenarios" or "erosion settings" to clarify

**It has been clarified by "cases regarding spatial and temporal settings" (Line 111).**

24. Row 106: space missing "(m).In"

**A space has been added (Line 149).**

25. Row 126: capitalize CENTURY model

**It has been revised as suggested (Line 176).**

26. Row 132: superscript missing in "ha-1"

**It has been replaced to % as suggested by the other reviewer (Line 183).**

27. Row 140: "We used discrimination ratio to", add "a" to "used a discrimination"

**"a" has been added (Line 190).**

28. Row 179: "At the meantime", replace to "In the meantime" or change to other expression

**"At the meantime" has been replaced with "In the meantime" (Line 229).**

29. Row 182-184: "For all the soil profiles, the component pools of each C isotopes of every layer are updated by homogeneously mixing the component materials every time step." – check grammar

**We have change "isotopes" to "isotope" (Line 233).**

30. Row 188: missing space after ";"

**This issue has been solved in the reference manage tool.**

31. Row 200: "$_{137}$Cs originates from bomb experiments between 1950 and 1970." Very simplified, it is worth clarifying that in the environment $_{137}$Cs concentrations are artificial fallout products from nuclear tests and reactor incidents, such as Chernobyl and Fukushima.

**It has been revised to make it more detailed as suggested (Line 250).**

32. Row 200: "It falls to the Earth's surface", would use other expression

**It has been revised (Line 251).**

33. Row 203: "The model reads the values", would remind the reader by clarifying which model

**The name of the model has been added (Line 254).**

34. Row 211: "the model was develop using", check grammar

**"develop" has been changed to "developed" (Line 265).**

35. Row 211: "complied", do you mean "compiled"?

**"complied" has been replaced with "compiled" (Line 265).**

36. Row 276: "replacement of lost at the" lost SOC?

**The missing "SOC" has been added (Line 400).**

37. Row 280: replace "negelation"

**"negelation" has been replaced with "neglection" (Line 408).**

38. Row 286: "from plant." Check grammar

**It has been changed to "from plants" (Line 415).**

39. Row 288-290: "At the same depth, soil profile of low soil advection and diffusion rate contains more degraded SOC than profile of high soil advetion and diffusion rate, and therefore soil profile of low soil advection and diffusion rate has less negative δ13C values." Check grammar, spelling and clarify the meaning of this sentence

**We have rewritten the sentence to make it clearer (Lines 417-418).**

40. Row 293-294: Check grammar and spelling

**It has been revised by using the plurals of "observation" and "cropland" (Line 424).**

41. Row 303: Check spelling

**"rate" has been replaced with "rates" (Line 433).**

42. Row 304-307: "Similar to δ13C profiles, erosion and deposition also have a truncation or burial effect of on the Δ14C profile and this results in the simulation that the eroding soil profiles have more negative Δ14C values compared to the stable soil profile while the profiles at the depositional sites have less negative Δ14C values than the stable soil profile (Figure 5c)." Check grammar and clarify the meaning

**We have revised the sentence to make it clearer (Lines 435-436).**

43. Row 318: "three-dimention" check spelling

**It has been changed to "three-dimension" (Line 449).**

44. Row 322: "This allows the model to be applied in various scenarios by setting relevant parameter values." Can you clarify which scenarios, e.g. different rates of erosion, different ranges of precipitation or change in vegetation?

**The types of scenarios have been clarified (Lines 461-462).**

45. Row 326: "The arrange", check expression/word

**It has been changed to "The arrangement" (Line 465).**

46. Row 330: "reprent", check spelling

**It has been replaced with "represent" (Line 469).**

47. Row n334: "a 3 pool" be consistent with using words vs. numbers, earlier it has been called a three-pool model

**It has been placed with "a three-pool" (Line 473).**

48. Row 339: "while depositional", check grammar "while the depositional"

**"the" has been added (Line 478).**

49. Row 345-346: Check grammar

**"causes" has been changed to "cause" (Line 485).**

50. Row 349-350: The link to the code appears to be broken.

**The files have been updated as suggested by the editor.**

**Reviewer 2:**

The authors developed a soil carbon model with coupled processes of decomposition, advection-diffusion and erosion-deposition. The model includes all carbon isotopes and 137Cs. It is a great effort to include all the carbon isotopes in the model. I have several suggestions and concerns below hoping to improve the manuscript.

1.Lack of technical details in the methods section is obvious. To name a few: how did the authors implement plant type change (changes in input, roots and/or else?) how did the authors test Suess effect in the model?

**A table on the description of various scenarios and how they were performed has been added (Table 2).**

How did the authors simulate spatial variability?

**The spatial variability is related to soil redistribution at the landscape scale. A paragraph has been added to describe in detail how the routing of runoff and soil particles are simulated in the model (Lines 142-145).**

How did the authors determine the initial conditions of the model?

**A detailed description of the procedure to estimate/set the initial profiles of $^{137}$Cs, C pools and C isotopic compositions has been added (Lines 270-280). We believe that this now provides sufficient information for the readers.**

What are the depth and depth intervals of the model?

**These are defined by parameters in the model code. Annotations have been added in the R script (reference_scenario.R) to explain the meanings of variables used in the codes. The R script file has been updated.**

More descriptions are necessary for 137Cs dynamics, such as equations and parameter values.

**Section 2.2.1 has been revised to include the lateral fluxes of $^{137}$Cs due to soil erosion. The decay of $^{137}$Cs (Eq. 21) has been added in section 2.2.5 (Lines 258-260).**

2.A data-model comparison is necessary for model evaluation. I suggest a direct comparison between model outputs and data in figure 7 and figure 8. For example, plot them together.

**Data presented in Figure 7 and Figure 8 (of last version) have been used for the model calibration together with C content data at these two study sites. The results derived from model calibration have been presented in Figure 1 and Figure 2 in the revised manuscript. Changes have also been made in Methods (Lines 113-132, Lines 291-317), Results (Lines 336-342), and Discussion (Lines 452-453) sections.**

3.Routine modeling activities such as sensitivity and uncertainty analysis are needed for model evaluations. Relative importance of the three main processes (decomposition, advection-diffusion, and erosion-deposition) need to be shown in some way.

**Thank you for this valuable suggestion. In response to this comment, the Fourier Amplitude Sensitivity Test (FAST) has been applied to the model to explore the importance of C decomposition, advection-diffusion and erosion-deposition in controlling C, $\delta^{13}$C and $\Delta^{14}$C profiles. The results is presented in Figure 7. Changes have also been made in Methods (Lines 318-327), Results (Lines 366-377), Discussion (Lines 454-459) and Conclusion (Lines 486-488) sections.**

4.An introduction and discussion of progress in model development in carbon isotopes would be very relevant.

**A paragraph on a review of progress in model development in carbon isotopes have been added (Lines 91-102).**

5.The results could use some more work. For example, I would not use current figure 1 as the first figure. It is not your central figure. I would first show some figures in model-data comparisons.

**Figure 1 has been removed from the revised manuscript. Figures on model-data (Figures 1 and 2) has been shown first as suggested (see the reply to Point 2 above).**

A few more comments:

1. What is WATEM_C short for? I did not find its full name throughout the text.

**The abbreviation has been explained (Line 136).**

2. Lines 100-105: L and S are slope steep and length factors, or the other way around?

**It is the other way around, and it has been corrected (Line 145).**

3. Line 211: developED

**It has been corrected as suggested (Line 265).**

4. Would a table be helpful for presenting all the modeling scenarios?

**A table including descriptions and implementation of model scenarios has been added (Table 2). Relevant revisions has also been done in the text (Line 282).**

5. A, S, and P in equations 7, 8, and 9: are not they carbon concentration (in the unit of for example, gC/m3) instead of carbon content? Because the authors model them with explicit depth. Please clarify.

**The reviewer is correct that A, S and P should be the content of various C pools. Similarly, the unit of C input into a given depth has been changed from Mg C ha $^{-1}$ yr$^{-1}$ to Mg C yr$^{-1}$ (Lines 182, and 183).**

6. Equation18: the terms on the right side are supposed to be partial differentials and K(z) is supposed to be inside the second-order differential due to the fact that K changes with z.

**Eq. 18 has been revised as suggested.**

7. what are the K values for Fig.1c? and similar issues for other relevant figures.

**Values of relevant parameters have been added in the figures (Figures 3-6 in the revised manuscript). We also extended Table 1 and added Tables 2 and 3 to display more information on parameter values of the model.**

8.I am surprised to see lack of depth dependence of 13C in Scenario 1. (Fig. 3a). Could the slower decomposition and lower carbon input along depth result in changes in 13C with depth, like 12C and 14C? Please clarify.

**Figure 6a (in the revised manuscript) shows the vertical variation of δ$^{13}$C values rather $^{13}$C content. In Scenario 1, slower decomposition and lower carbon input along depth result in changes in 13C with depth, but 12C has similar changes with soil depth, and therefore the 13C composition (δ$^{13}$C) does not change with soil depth. A sentence has been added to explain this (Line 409).**

9. Fig. 6 is difficult to read. I'd suggest the authors use colored scheme.

**The figure has been changed to color maps (Figure 8 in the revised version).**

10. Line 280: negligence instead of negelation

**It has been replaced with neglection (Line 408).**

---

## Author Response (AR2)

Dear editor,

Thank you very much for your review on our manuscript. With respect to the concerns on test data and archive files of the executive editor, we have used Zenodo to archive the model manual, test data and examples of using the model. Please find the details in code and data availability section (Lines 465-468).

Yours sincerely,

The authors

Topical Editor Decision: Publish subject to minor revisions (review by editor) (10 Aug 2020) by Sandra Arndt

Comments to the Author:

Dear Dr. Wang,

thank you for submitting a revised version of your manuscript and a response to the two detailed reviewer reports. The reviewers had expressed several concerns regarding the reproducibility and representation of your study and you have addressed most of these concerns in the revised version.

However, I would like to invite you to also address the executive editor's comment regarding data source and code archiving before I can recommend publication. Specifically, see exec. editor comment:

"...the datasets used to conduct the evaluation experiments presented must be cited from the code and data availability section with enough precision to allow a reader to reproduce the work in the manuscript."

" Github is an excellent development platform, but it lacks the features required of an archive. GitHub themselves tell authors to use Zenodo for this purpose. The authors should follow the procedure detailed there to archive the exact version of the software used to create the results presented: https://guides.github.com/activities/citable-code/. The resulting Zenodo repositories present the bibliography entries to use"

Please refer to the following documents containing

- further details on code and data availability requirements: https://www.geoscientific-model-development.net/about/code_and_data_policy.html

and its motivation:

https://doi.org/10.5194/gmd-12-2215-2019

Best, Sandra Arndt